# Causal roles of prefrontal cortex during spontaneous perceptual switching are determined by brain state dynamics

Takamitsu Watanabe[1,2]*

[1]International Research Centre for Neurointelligence, The University of Tokyo Institutes for Advanced Study, Tokyo, Japan; [2]RIKEN Centre for Brain Science, Saitama, Japan

**Abstract** The prefrontal cortex (PFC) is thought to orchestrate cognitive dynamics. However, in tests of bistable visual perception, no direct evidence supporting such presumable causal roles of the PFC has been reported except for a recent work. Here, using a novel brain-state-dependent neural stimulation system, we identified causal effects on percept dynamics in three PFC activities— right frontal eye fields, dorsolateral PFC (DLPFC), and inferior frontal cortex (IFC). The causality is behaviourally detectable only when we track brain state dynamics and modulate the PFC activity in brain-state-/state-history-dependent manners. The behavioural effects are underpinned by transient neural changes in the brain state dynamics, and such neural effects are quantitatively explainable by structural transformations of the hypothetical energy landscapes. Moreover, these findings indicate distinct functions of the three PFC areas: in particular, the DLPFC enhances the integration of two PFC-active brain states, whereas IFC promotes the functional segregation between them. This work resolves the controversy over the PFC roles in spontaneous perceptual switching and underlines brain state dynamics in fine investigations of brain-behaviour causality.

## Editor's evaluation

This single-author work, by combining real-time closed-loop EEG-TMS and sophisticated computational modelling to characterize ongoing brain states, impressively demonstrates the causal role and different functions of several prefrontal regions in modulating bistable perception, in a brain-state-dependent way.

*For correspondence: takamitsu.watanabe@ircn.jp

Competing interest: The author declares that no competing interests exist.

## Introduction

Dynamic and flexible changes are among the fundamental key properties of human cognition. Bistable visual perception has been widely used to investigate such cognitive dynamics (*Brascamp et al., 2018*; *Sterzer et al., 2009*), and the prefrontal cortex (PFC)—in particular, right frontal eye fields (FEF), dorsolateral PFC (DLPFC) and inferior frontal cortex (IFC)—is thought to be involved in the spontaneous perceptual switching (*Brascamp et al., 2018*; *Kleinschmidt et al., 1998*; *Lumer et al., 1998*; *Lumer and Rees, 1999*; *Panagiotaropoulos et al., 2020*; *Panagiotaropoulos et al., 2012*; *Sterzer et al., 2009*; *Sterzer et al., 2002*; *Wang et al., 2013*; *Weilnhammer et al., 2013*). Theoretical work also indicates that top-down signals from the PFC to the visual cortex are essential to perceptual inference (*Hohwy et al., 2008*; *Weilnhammer et al., 2017*). These studies imply that inhibitory neural modulation of the PFC should induce behavioural changes in the bistable visual perception.

However, no empirical human study has identified such behavioural causality of the PFC (*Brascamp et al., 2018*) except for a recent work administering theta-burst transcranial magnetic stimulation

**eLife digest** A cube that seems to shift its spatial arrangement as you keep looking; the elegant silhouette of a pirouetting dancer, which starts to spin in the opposite direction the more you stare at it; an illustration that shows two profiles – or is it a vase? These optical illusions are examples of bistable visual perception. Beyond their entertaining aspect, they provide a way for scientists to explore the dynamics of human consciousness, and the neural regions involved in this process.

Some studies show that bistable visual perception is associated with the activation of the prefrontal cortex, a brain area involved in complex cognitive processes. However, it is unclear whether this region is required for the illusions to emerge. Some research has showed that even if sections of the prefrontal cortex are temporally deactivated, participants can still experience the illusions.

Instead, Takamitsu Watanabe proposes that bistable visual perception is a process tied to dynamic brain states – that is, that distinct regions of the prefontal cortex are required for this fluctuating visual awareness, depending on the state of the whole brain. Such causal link cannot be observed if brain activity is not tracked closely.

To investigate this, the brain states of 65 participants were recorded as individuals were experiencing the optical illusions; the activity of their various brain regions could therefore be mapped, and then areas of the prefrontal cortex could precisely be inhibited at the right time using transcranial magnetic stimulation. This revealed that, indeed, prefrontal cortex regions were necessary for bistable visual perception, but not in a simple way. Instead, which ones were required and when depended on activity dynamics taking place in the whole brain. Overall, these results indicate that monitoring brain states is necessary to better understand – and ultimately, control – the neural pathways underlying perception and behaviour.

(TMS) over the right IFC (*Weilnhammer et al., 2021*). Instead, a previous TMS study reported that neural suppression of the right DLPFC did not affect bistable visual perception (*de Graaf et al., 2011*). Other studies claimed that the PFC activity is not essential to the emergence of multistable perception (*Brascamp et al., 2015*; *Harrison and Tong, 2009*) but mere a consequence of it (*Block, 2020*; *Brascamp et al., 2015*; *Frässle et al., 2014*; *Knapen et al., 2011*).

Why is it so difficult to detect the prefrontal causality in the multistable perception? Here, given that the whole-brain neural activity during bistable perception is described as large-scale brain state dynamics (*Watanabe et al., 2014c*), we hypothesise that causal roles of the PFC should also be dynamically changing during the fluctuation of visual awareness. That is, if the neural activity in the multistable perception can be stated as dwelling in and transitions between a parsimonious number of brain states (*Watanabe et al., 2014c*), the detectability of the prefrontal causal effects on the perceptual awareness should be determined by the brain state in which the neural activity pattern is staying when the neural stimulation is administered. If so, such state-dependent behavioural causality should be hardly observed when we intervene in the PFC activity without tracking the brain state dynamics.

We directly tested this hypothesis with a brain-state-dependent neural stimulation system (*Bergmann, 2018*; *Silvanto et al., 2008*; *Zrenner et al., 2016*). The system was devised by linking energy landscape analysis (*Ezaki et al., 2017*; *Gu et al., 2018*; *Kang et al., 2017*; *Watanabe et al., 2014c*; *Watanabe and Rees, 2017*)—a data-driven method to identify the neural dynamics during seemingly random behaviours—to an electroencephalogram (EEG)-triggered TMS (*Bergmann et al., 2016*; *Schaworonkow et al., 2019*; *Stefanou et al., 2018*; *Zrenner et al., 2018*).

To focus on the higher order cortex, we did not adopt a test of binocular rivalry, in which the perceptual fluctuation has often been linked with the neural activity in the lower level brain systems such as the visual cortex (*Haynes et al., 2005*; *Lee et al., 2005*; *Leopold and Logothetis, 1996*; *Meng and Tong, 2004*; *Miller et al., 2000*; *Pettigrew and Miller, 1998*; *Polonsky et al., 2000*). Instead, we used a test of bistable visual perception induced by a structure-from-motion (SFM) stimulus (*Figure 1a*), in which the same visual stimulus is presented to both the eyes of participants, and the higher order cortex is considered to be more involved in the perceptual fluctuation (*Brascamp et al., 2018*; *Knapen et al., 2011*; *Meng and Tong, 2004*).

**Figure 1.** Experiment design. While the participants were presented with a structure-from-motion (SFM) stimulus (**a**), we recorded EEG signals from the seven brain regions (**b**). DLPFC, dorsolateral prefrontal cortex; IFC, inferior frontal cortex; FEF, frontal eye fields; aSPL/pSPL, anterior/posterior superior parietal lobule; LOC, lateral occipital complex. In the control experiments, we first conducted offline energy landscape analysis, whose results allowed us to categorise brain activity patterns into either of the three major brain states (**c**, left). By implementing such classification information to an online EEG analysis, we tracked brain state dynamics and administered inhibitory TMS in state-/state-history-dependent manners (**c**, right).

## Results

### Validation of energy landscape analysis in EEG

As a preparation, we examined whether the current EEG system captured qualitatively the same brain state dynamics as those found in our previous fMRI study employing the same SFM stimulus (*Watanabe et al., 2014c*). To this end, we recorded gamma-band EEG signals from the seven cortical regions (*Watanabe et al., 2014c*; *Figure 1b*) of 65 healthy adults while they were experiencing the SFM-induced bistable visual perception (left panel of *Figure 1c*) and applied an offline energy landscape analysis to the data (Control Experiment I).

First, we confirmed that a pairwise maximum entropy model—a basis of the energy landscape analysis—was well fitted to the data in all the participants (fitting accuracy >84 %; *Figure 2a*). Based on the model, we then found that all the participants had almost the same energy landscape structure consisting of three major brain states: Frontal-area-dominantly-active state (F state), Visual-area-dominantly-active state (V state) and Intermediate state (Int state) (e.g. *Figure 2b* for Participant 1). As in our previous fMRI work, these three states were significantly separated and independent from each other (i.e. the energy barrier >1; *Figure 2c*) and dominated almost all the energy surfaces (*Figure 2d*).

Such model-based estimations were consistent with the actual EEG data. Even in the neural data, almost all the brain activity patterns could be classified into either of the three major brain states (the other states < 0.1 %; *Figure 2e*). The degree to which each of the three brain states occupied the energy landscape (i.e. the basin size) was significantly correlated with how frequently the brain state appeared in the EEG data ($r_{64}$ >0.57; Mean Average Percentage Error, MAPE, < 6.8 %; *Figure 2f*).

Moreover, the neural dynamics between these three major brain states were qualitatively equivalent to those seen in our fMRI study: there was almost no direct transition between F and V state ( < 0.4%; *Figure 2g,h*), and thus the brain activity pattern always used Int state as a stepping stone to travel between F and V state (*Figure 2i*).

Finally, we confirmed that, as shown in our fMRI work, the length of such neural travel between F and V state via Int sate was a good indicator to predict the individual behavioural differences in the percept duration ($r_{64}$ = 0.67; *Figure 2j*). Furthermore, this brain-behaviour association was preserved at an individual level: even within each participant, the length of the F-Int-V-Int-F travel calculated for each run accurately predicted the percept duration in the run (e.g. *Figure 2k* for Participant 1; for all the participants, $r_{19}$ >0.54, MAPE <7.2%, *Figure 2l,m*). Note that these energy landscape analyses used no behavioural information to identify the brain state dynamics; thus, the significant brain-behaviour correlations are not consequences of circular analysis.

In sum, these findings show that, at least with an offline analysis, the current EEG system can identify qualitatively the same brain state dynamics underpinning the SFM-induced bistable visual perception as seen in our previous fMRI study (*Watanabe et al., 2014c*).

### Validation of state-dependent neural stimulation

These results have sufficient information to categorise each neural activity pattern at each time-point into either of the three major states. In the Control Experiment II, we implemented such

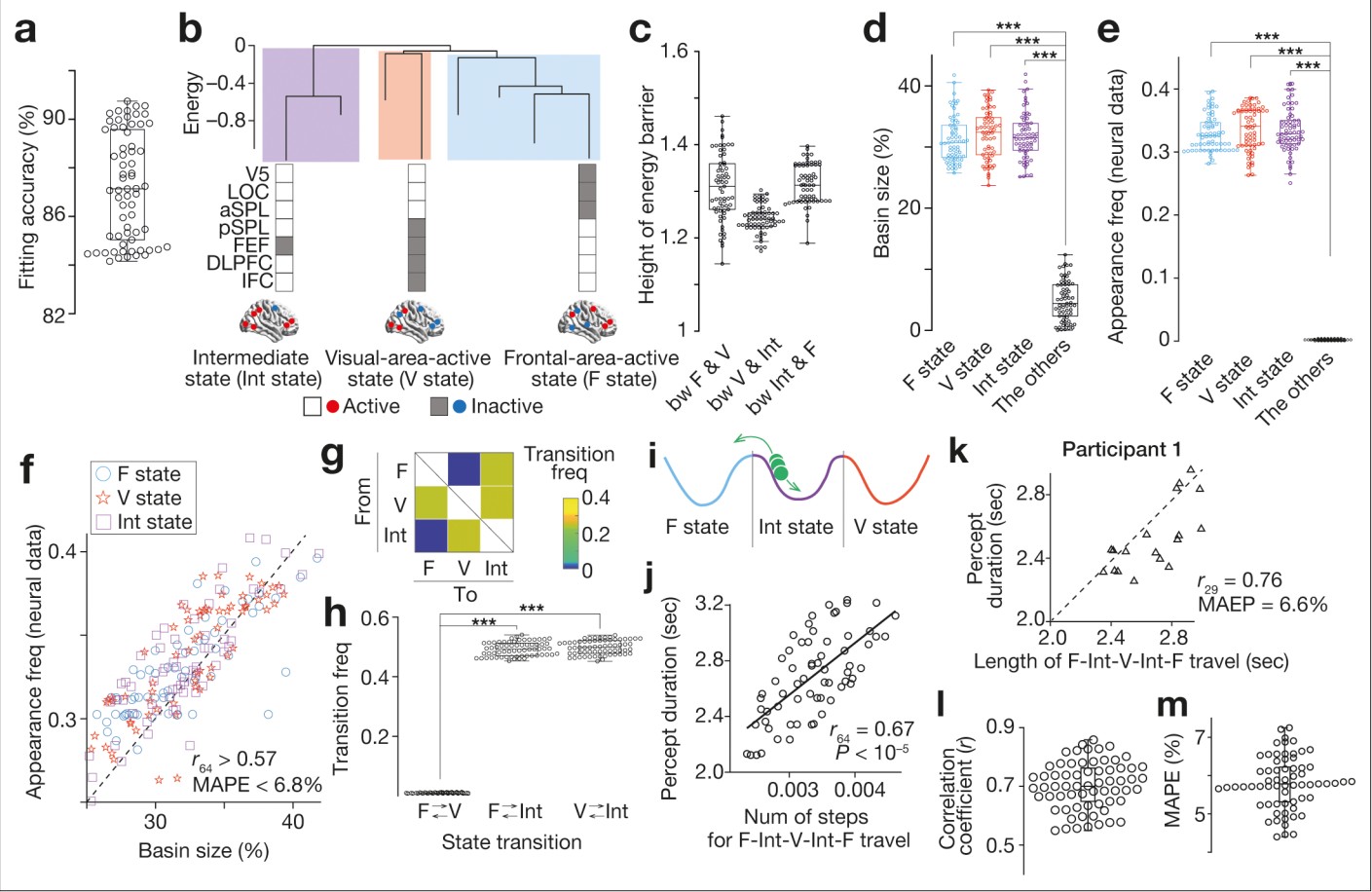

**Figure 2.** Feasibility of energy landscape analysis for EEG data. (**a-d**) We confirmed that the brain state dynamics found in the EEG data were qualitatively the same as those seen in our previous fMRI study (**Watanabe et al., 2014c**). A pairwise maximum entropy model, a basis for the energy landscape analysis, was accurately fitted to the EEG data (**a**) and yielded almost the same energy landscape structure with three major brain states (e.g. panel **b** for Participant 1). The three brain states were significantly separated with high energy barriers (**c**) and dominated almost all the energy surface (**d**). (**e and f**) These estimations based on the energy landscape structure were consistent with the actual neural data. In the EEG data, almost all the neural activity patterns could be categorised into either of the three major brain states (**e**). The basin sizes in the model accurately predicted the appearance frequency of the major brain states (**f**; each mark represents each individual). (**g–j**) We then performed random-walk simulations on the energy landscape and found that there was no direct transition between F and V state (panel **g** for the group average of the transition matrix; panel **h** for the transition frequencies for all the participants). This result suggests that, during the bistable visual perception, the neural activity pattern (a green ball in panel **i**) is travelling between F and V state via Int state (**i**). As shown in our fMRI work, the number of steps to complete this F-Int-V-Int-F travel was correlated with the behavioural percept duration across participants (**j**). (**k–m**). We confirmed that this brain-behaviour correlation was seen at an individual level. Technically, for each run in a participant, we calculated the length of the neural travel between F and V state via Int state and compared it with the percept duration for the run. Panel **k** shows such a comparison in Participant 1, indicating a significant correlation between the brain state dynamics and perceptual stability in the participant. In the other participants, these associations were preserved (panel **l** for the correlation coefficients; panel **m** for the mean average per cent error, MAPE). In the panels **a**, **c-e**, **h**, **j**, **l** and **m**, dots represent participants. ***, $P < 10^{-3}$ in a paired $t$-test ($df = 64$).

classification information into an online EEG analysis (**Stefanou et al., 2018**; **Zrenner et al., 2018**; right panel of **Figure 1c**) and examined whether the online analysis could track the bran state dynamics accurately and enable us to administer TMS while the brain activity pattern was dwelling in a specific brain state.

First, we confirmed that the brain state identification in the online analysis was significantly similar to that obtained by the offline analysis (similarity >82.3 % for the three major brain states; **Figure 3a**). Then, by linking this online EEG signal processing to monophasic TMS, we succeeded in triggering a burst of inhibitory TMS only when the neural activity pattern was staying in a specific major brain state (accuracy >84.8%, **Figure 3b**; latency <0.8 ms, **Figure 3c**).

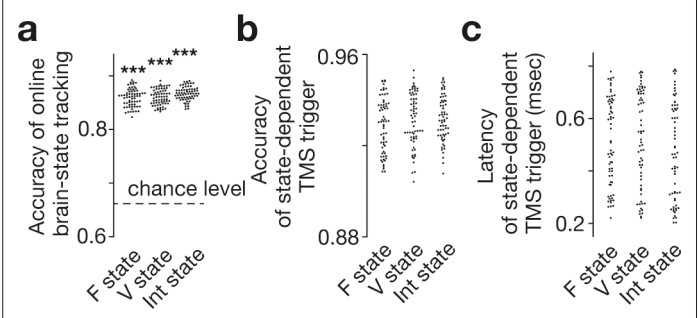

**Figure 3.** Accuracy of online brain state tracking and triggering of TMS. The online EEG analysis tracked brain state dynamics as accurately as the offline analysis did (**a**). Brain-state-dependent TMS triggering system also achieved accurate neural tracking (**b**) with short delays (**c**). Note that the accuracy in the TMS trigger (**b**) tended to be higher than that in simple brain-state tracking (**a**), presumably because TMS was triggered only when a specific brain state was expected to continue in a certain period (here, 150 ms) and resultantly reduce the rate of false negative. Also, such a criterion should work as temporal smoothing and improve the signal-to-noise ratio. Each dot represents each participant. *** indicates $P_{\text{Bonferroni}} <0.001$ in paired $t$-tests (df = 64).

## State-dependent causality

By applying this state-dependent TMS over the three PFC regions (i.e. DLPFC, IFC and FEF; *Figure 4*), we found that the three prefrontal areas had different causal behavioural effects on the spontaneous perceptual switching in a brain-state-dependent manner (N = 34; $F_{8,33}=25.9$, p < 10$^{-3}$ for the main effect in a two-way ANOVA; *Figure 4b*).

The neural inhibition of DLPFC during F state prolonged the percept duration ($t_{33} = 9.4$, $P_{\text{Bonferroni}} < 0.001$ in a post-hoc paired $t$-test, Cohen's $d = 1.7$), whereas that during V or Int state induced no behavioural change ($t_{33} <1.7$, $P_{\text{Bonferroni}} > 0.05$, $d < 0.3$).

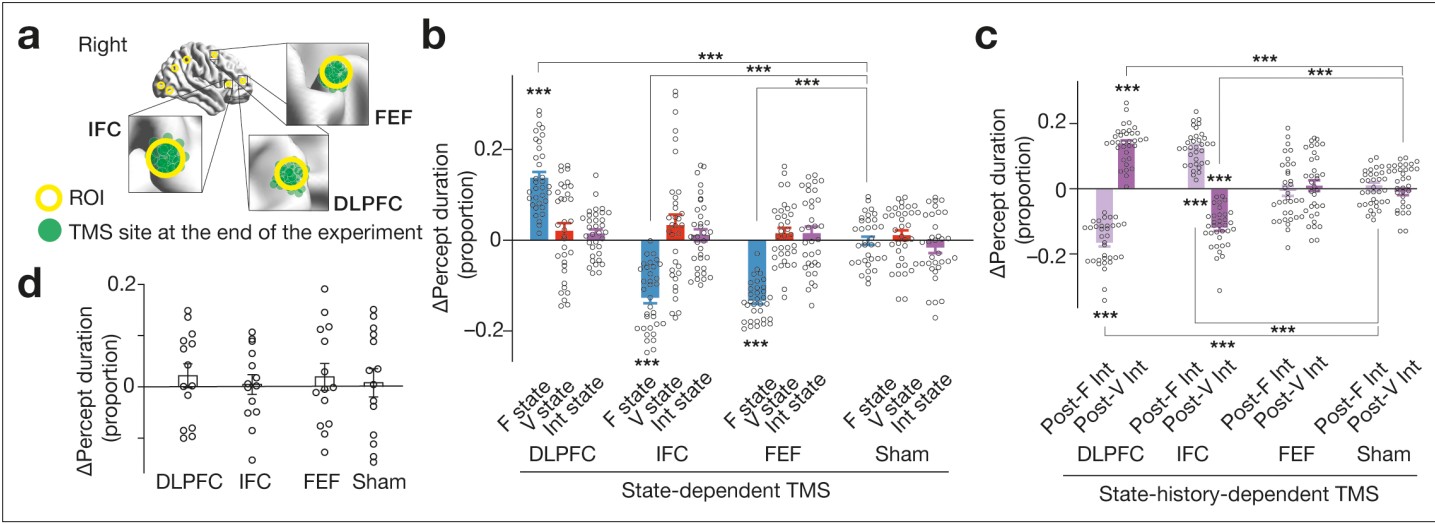

**Figure 4.** Behavioural results . (**a**) We administered inhibitory TMS over the three PFC regions. In one TMS condition, we placed the TMS coil over one of the PFC areas using a stereoscopic neuro-navigation system based on the MNI coordinates at the beginning of each experiment day. At the end of the day, we re-measured the coordinates of the TMS coil using the navigation system. Finally, we averaged the coordinates across the 4-day sessions in the main experiment. The green circles represent such mean MNI coordinates of the stimulated brain site for each participant. The green circles were mapped closely onto the original coordinates (the centres of the yellow circles). (**b**) State-dependent TMS revealed that causal behavioural roles of the three PFC regions are detectable when the brain activity pattern stays in F state. The Y-axis shows the proportional changes in the median percept duration ( = (TMS–control)/control). The X-axis indicates the TMS conditions. *** indicates $P_{\text{Bonferroni}} <0.001$ in paired $t$-tests (df = 33). Each circle represents each participant. The error bars show the s.e.m. (**c**) The DLPFC and IFC showed state-history-dependent behavioural causal effects on bistable visual perception, while FEF did not. *** indicates $P_{\text{Bonferroni}} <0.001$ in paired $t$-tests (df = 33). (**d**) No significant behavioural change was induced after stronger and longer TMS (30min quadripulse TMS with 90 % AMT) over the same three PFC regions. The error bars show the s.e.m.The intensity of the current TMS was set at 70 % AMT.

Regarding IFC, the F-state-dependent TMS destabilised the visual perception ($t_{33}$ = 10.4, $P_{Bonferroni}$ < 0.001, $d$ = 1.8), whilst neither V- nor Int-state-dependent TMS induced any behavioural effect ($t_{33}$ <1.3, $P_{Bonferroni}$ > 0.05, $d$ < 0.29).

The F-state-dependent TMS over FEF reduced the percept duration ($t_{33}$ = 10.5, $P_{Bonferroni}$ < 0.001, $d$ = 1.9), whereas no behavioural change was observed in the other FEF TMS conditions ($t_{33}$ <0.8, $P_{Bonferroni}$ > 0.05, $d$ < 0.39).

## State-history-dependent causality

If some behavioural causalities are detectable when the whole-brain neural activity pattern is dwelling in specific brain states, others may emerge when the neural activity pattern has finished travelling a particular brain state trajectory. We then tested this hypothesis and found such state-history-dependent causality ($F_{5,33}$=85.6, p < 10$^{-3}$ for the main effect in a two-way ANOVA; *Figure 4c*). Here, we focused on Int state because it is the sole brain state that had two incoming pathways (i.e. a path from F state and one from V state).

The TMS over DLPFC during Int state immediately after F state shortened the percept duration ($t_{33}$ = 12.9, $P_{Bonferroni}$ <0.001 in a post-hoc paired *t*-test, $d$ = 2.7), whereas the TMS over DLPFC during Int state right after V state prolonged it ($t_{33}$ = 13.3, $P_{Bonferroni}$ <0.001, $d$ = 2.3).

For IFC, the Post-F Int-state-dependent TMS enhanced the perceptual stability ($t_{33}$ = 9.1, $P_{Bonferroni}$ <0.001, $d$ = 1.6), whilst that in Post-V Int state weakened it ($t_{33}$ = 12.7, $P_{Bonferroni}$ <0.001, $d$ = 2.3).

No change in the percept duration was observed when we administered TMS over FEF during Post-F or Post-V Int state ($t_{33}$ <0.58, $P_{Bonferroni}$ > 0.05, $d$ < 0.19).

Given that no significant behavioural change was induced by longer and stronger TMS (here, 30 min quadripulse TMS)(*Hamada et al., 2009*; *Hamada et al., 2007*; *Watanabe et al., 2015*; *Watanabe et al., 2014a*) over the same PFC regions (N = 14; $F_{2,13}$=0.16, p = 0.85 for the main effect in a two-way ANOVA; *Figure 4d*), these results suggest that the causal behavioural roles of the PFC areas in the bistable visual perception become explicit and measurable only when we intervene in the neural activity in state-/state-history-dependent manners.

## Effects on Energy Landscape Structure

These brain-state-dependent behavioural responses imply that the underlying neural mechanisms could be accounted for by brain state dynamics. .

To reveal such brain mechanisms, we first formulated working hypotheses on the neural effects of TMS by conducting numerical simulation on how local brain inhibition would affect the energy landscape structure and, resultantly, the brain state dynamics. In particular, we calculated changes in the heights of the energy barriers between the three major brain states (*Figure 5a*), because the barrier

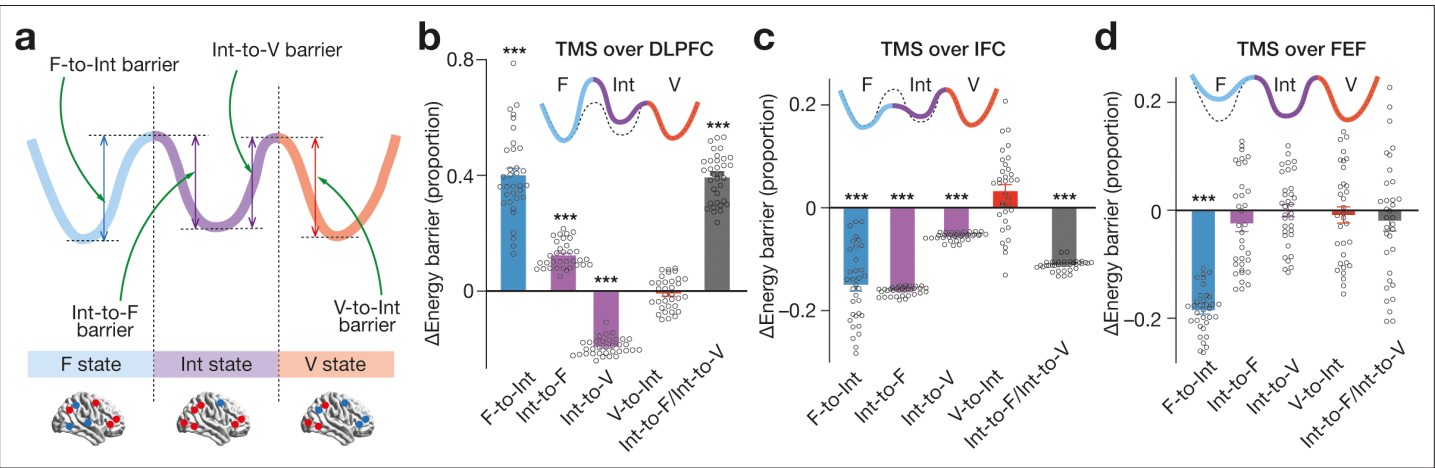

**Figure 5.** Effects on energy landscape structure . We numerically simulated how inhibitory TMS would affect the heights of the F-to-Int, Int-to-F, Int-to-V and V-to-Int barrier for each participant (**a**), which predicted significant changes in the energy barriers (panel **b** for TMS over DLPFC, panel **c** for TMS over IFC and panel **d** for TMS over FEF). In the panels **b-d**, the solid tricolour curves schematically represent post-TMS energy landscape structures, whereas the dashed curves denote the original energy landscape. *** indicates $P_{Bonferroni}$ <0.001 in paired *t*-tests (df = 33). The error bars show the s.e.m.

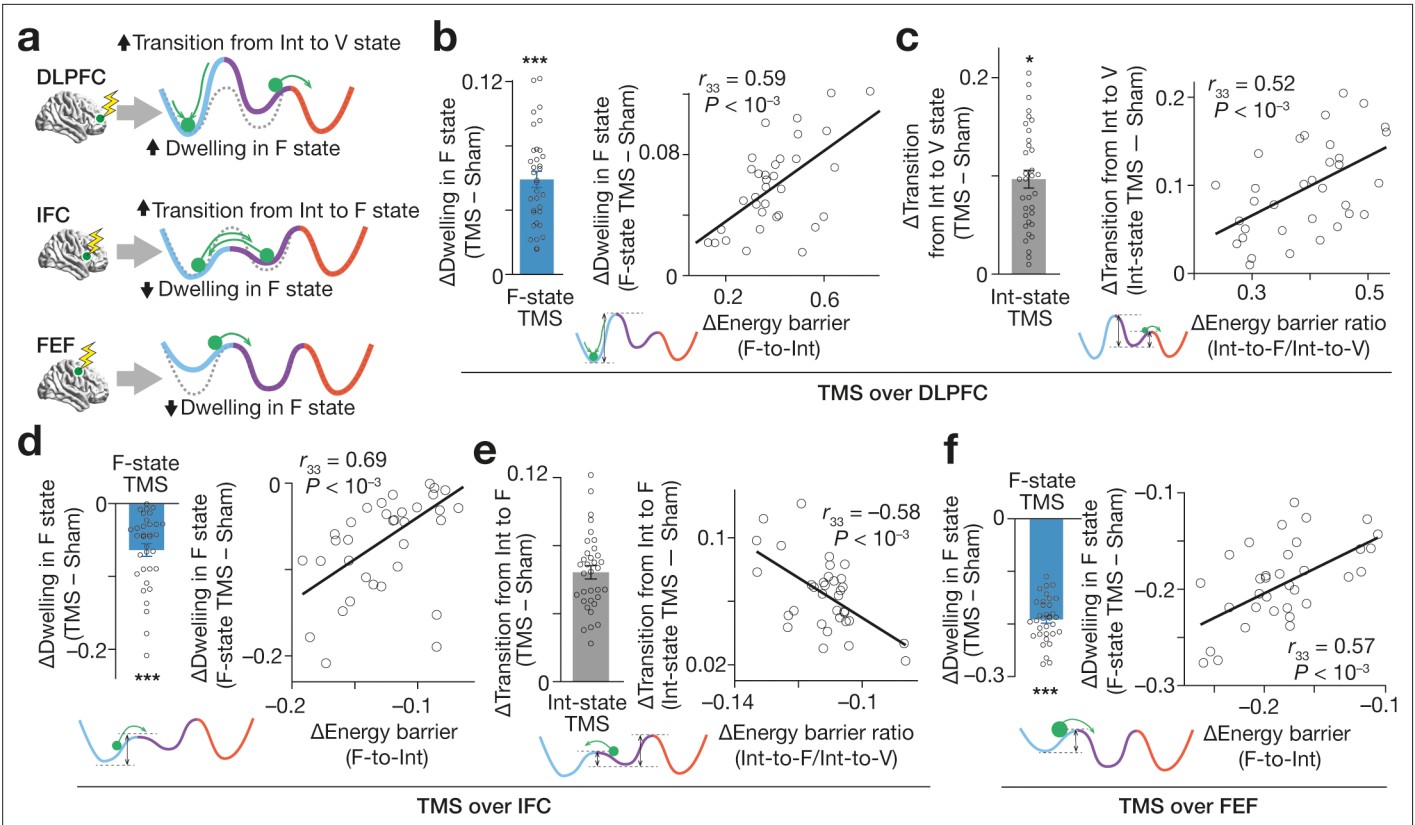

**Figure 6.** Effects on brain state dynamics. (**a**) The numerical simulation (*Figure 5*) enables us to infer TMS-induced effects on the brain state dynamics. The tricolour curves schematically indicate the energy landscapes modified by TMS, whereas the dashed curves show the original energy landscapes. The green circles show the brain activity pattern. (**b–f**) These inferences were empirically confirmed. The X-axes show the changes in the hypothetical energy barrier heights, whereas the Y axes indicate TMS-induced neural effects on the brain state dynamics. The circles represent the participants. *** and * indicate $P_{Bonferroni}$ <0.001 and $P_{Bonferroni}$ <0.05 in paired *t*-tests (*df* = 33), respectively. The error bars show the s.e.m.

heights are associated with the dwelling time in the brain states and inversely correlated with the transition frequency between them (*Watanabe et al., 2014c*).

As to DLPFC (*Figure 5b*), the numerical simulation showed that the TMS should increase the energy barrier heights between F and Int state ($F_{4,33}$=340.1, p < $10^{-3}$ for the main effect in a two-way ANOVA; $t_{33}$ >15.2, $P_{Bonferroni}$ <0.001 in post-hoc paired *t*-tests, *d* > 2.6) and decrease the barrier height from Int to V state ($t_{33}$ = 38.0, $P_{Bonferroni}$ <0.001, *d* = 6.7). As a result, the ratio of the Int-to-F barrier to the Int-to-V barrier should significantly increase ($t_{33}$ = 27.1, $P_{Bonferroni}$ <0.001, *d* = 4.8).

Regarding IFC (*Figure 5c*), the neural inhibition of the region should alleviate the energy barriers between F and Int state ($F_{4,33}$=89.10, p < $10^{-3}$ for the main effect in a two-way ANOVA; $t_{33}$ >11.6, $P_{Bonferroni}$ <0.001, *d* > 2.0) and that from Int to V state ($t_{33}$ = 48.9, $P_{Bonferroni}$ <0.001, *d* = 8.6). The magnitude of the decrease in the Int-to-F barrier height should become larger than that in the Int-to-V barrier, which would result in a significant decrease in the ratio of the Int-to-F barrier to the Int-to-V barrier ($t_{33}$ = 61.2, $P_{Bonferroni}$ <0.001, *d* = 10.7).

For FEF (*Figure 5d*), its neural suppression should lower the energy barrier from F to Int state ($F_{4,33}$=32.4, p < $10^{-3}$ for the main effect in a two-way ANOVA; $t_{33}$=25.3, $P_{Bonferroni}$ < 0.001, *d* = 4.4) but induce no significant change in the other barriers ($t_{33}$ <1.7, $P_{Bonferroni}$ > 0.05, *d* < 0.27).

## Effects on brain state dynamics

These structural changes in the energy landscapes indicate how the TMS affected the brain state dynamics (*Figure 6a*).

For example, the higher F-to-Int energy barrier—which is predicted to occur after TMS over DLPFC (*Figure 5b*)—would enhance the segregation between F and Int state, impede the transition from F

to Int state and prolong the dwelling in F state. Moreover, such longer F-state dwelling should be observed in F-state-dependent TMS condition the most clearly.

By the same logic, the relatively lower Int-to-V barrier, which is also expected to occur after TMS over DLPFC, should increase the Int-to-V transitions. Regarding the TMS over IFC, the lower F-to-Int energy barrier should shorten the F-state dwelling, whereas the relatively lower Int-to-F barrier should increase the Int-to-F transitions. In the TMS-over-FEF condition, the lower F-to-Int barrier should reduce the F-state dwelling.

We tested and confirmed these hypotheses by measuring the dwelling time of the three major brain states and transition frequencies between them for all the state-dependent TMS conditions.

In the experiments administering TMS over DLPFC, the increase in the F-to-Int energy barrier was correlated with the longer F-state dwelling seen in the F-state-dependent TMS ($t_{33}$ = 11.6, $P_{Bonferroni}$ <0.001, $d$ = 2.0; $r_{33}$ = 0.59, p < 0.001; **Figure 6b**). Also, the higher Int-to-F energy barrier was associated with more frequent Int-to-V transitions that was seen in the Int-state-dependent TMS ($t_{33}$ = 3.3, $P_{Bonferroni}$ <0.05, $d$ = 1.8; $r_{33}$ = 0.52, p < 0.001; **Figure 6c**).

In the sessions administering TMS over IFC, the lower F-to-I energy barrier was correlated with the shorter F-state dwelling that was measured in the F-state-dependent neural suppression ($t_{33}$ = 6.9, $P_{Bonferroni}$ <0.001, $d$ = 1.2; $r_{33}$ = 0.69, p < 0.001; **Figure 6d**). The lower Int-to-F barrier accurately predicted more frequent Int-to-F transitions ($t_{33}$ = 13.0, $P_{Bonferroni}$ <0.001, $d$ = 2.8; $r_{33}$=–0.58, p < 0.001; **Figure 6e**) that were seen in the Int-state-dependent TMS condition.

In the TMS-over-FEF conditions, the lower F-to-Int barrier was associated with the shorter F-state dwelling that was observed in the F-state-dependent TMS session ($t_{33}$ = 26.1, $P_{Bonferroni}$ <0.001, $d$ = 4.8; $r_{33}$ = 0.57, p < 0.001; **Figure 6f**).

These results clarify TMS-induced effects on the brain state dynamics during the bistable visual perception and demonstrate that such neural effects are underpinned by the structural changes in the energy landscapes.

## Temporal decay of neural effects

How long did such neural effects continue after each TMS? To answer this question, we tracked the magnitudes of the correlations between the empirical neural effects and the numerically-calculated energy barrier changes (**Figure 6b,c,d,e,f**) when we were sliding the time window that was used to empirically quantify such neural effects (**Figure 7a**). This analysis detected that all the correlations began to weaken approximately 1.5 s after each TMS (**Figure 7b**). This result indicates that the current TMS-induced neural effects started to decay within ~1.5 s after the stimulation, and the brain state dynamics began to return to the original forms in such a time scale.

## Neural mechanisms behind behavioural causality

Finally, we evaluated associations between these behavioural, numerical and neural responses by mediation analysis and found that the causal behavioural effects are induced by transient changes in the brain state dynamics and attributable to structural changes in the energy landscapes ($P$ < 0.05 for all the indirect effects, $\alpha \times \beta$; **Figure 8**).

As to the state-dependent behavioural changes, the longer percept duration seen after the F-state-dependent TMS over DLPFC was largely due to the prolonged F-state dwelling, which was induced by the higher F-to-Int energy barrier (**Figure 8a**). In contrast, the de-stabilisation of visual perception seen after the

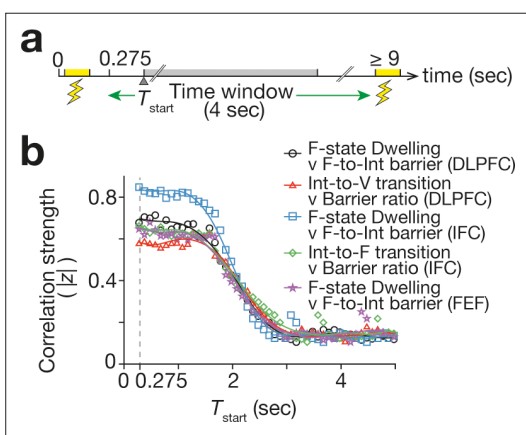

**Figure 7.** Duration of TMS effects. To assess the duration of the TMS-induced neural effects, we examined how the correlations seen in **Figure 6** changed when we slid the 4 s time window that was used to measure the neural effects. In all the correlations, the correlation strength (here, the absolute value of the Fisher-transformed correlation coefficient, |z|) began to decay ~1.5 s after the TMS. The X-axis shows the start timing of the 4 s time window ($T_{start}$).

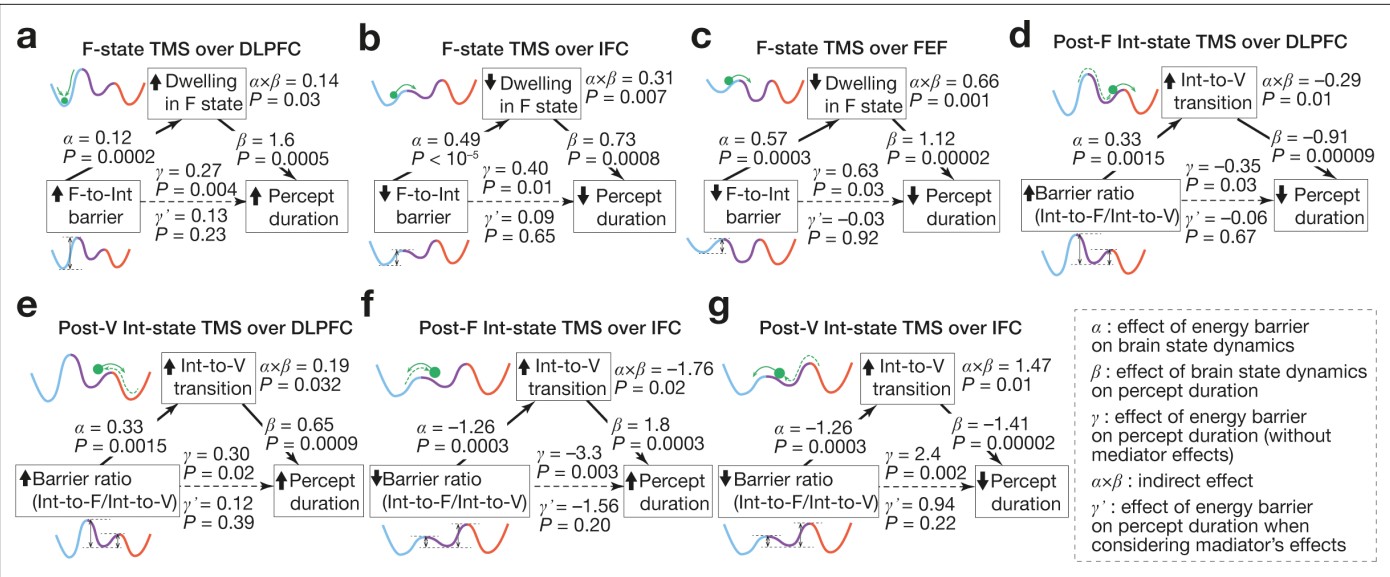

**Figure 8.** Associations between energy landscape structure, brain state dynamics and behaviour. The mediation analyses showed that the structural changes in the energy landscapes affect the brain state dynamics, which results in the causal behavioural responses. The statistical significance of the $\alpha$, $\beta$, and $\gamma$ validate our application of the mediation analysis to the current data. The statistical significance of the $\alpha \times \beta$ and the insignificance of the $\gamma'$ support the conclusions.

F-state-dependent TMS over IFC/FEF was caused by the shorter F-state dwelling, which was attributable to the lower F-to-Int energy barrier (**Figure 8b,c**).

The state-history-dependent behavioural causality could also be seen as a consequence of the transient changes in the brain state dynamics, given that the percept duration is closely linked with the length of the F-Int-V-Int-F travel (**Figure 2j,k**): According to the mediation analysis, the shorter percept duration seen after the Post-F Int-state-dependent TMS over DLPFC was caused by the increase in the Int-to-V transitions, which was triggered by the larger gap between the Int-to-F barrier and Int-to-V barrier (**Figure 8d**). This statistical result is reasonable because the relatively lower Int-to-V energy barrier would facilitate the Int-to-V transitions transiently, accelerate the completion of the F-Int-V travel and shorten the percept duration.

Conversely, the longer percept duration yielded by the Post-V Int-state-dependent TMS over DLPFC is interpretable as a result of the relatively higher Int-to-F energy barrier's impeding the Int-to-F transitions, accelerating the backward moves to V state and slowing down the completion of the F-Int-V travel (**Figure 8e**).

The mediation analysis showed that the state-history-dependent causal roles of IFC were also accountable by the same logic. The longer percept duration yielded by the TMS over IFC during Post-F Int state could be regarded as a behavioural manifestation of the slowdown of the F-Int-V travel due to the more frequent backward Int-to-F transitions, which was originated from the relatively lower Int-to-F energy barrier (**Figure 8f**). In contrast, the shorter percept duration induced by the TMS over IFC during Post-V Int state can

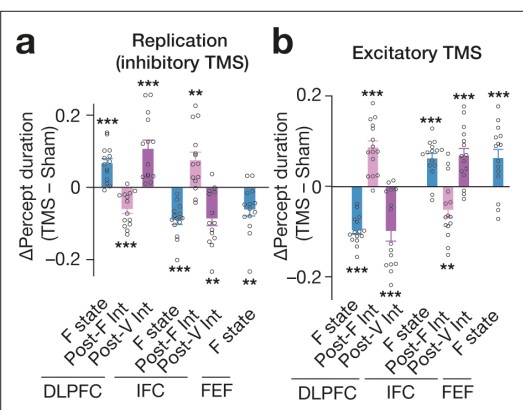

**Figure 9.** Validation tests. (**a**) All the main behavioural causal responses were replicated in an independent cohort (N = 14). (**b**) In another independent experiment (N = 15), we administered excitatory TMS over the same PFC regions. The excitatory stimulaton induced behavioural effects opposite to those seen in the inhibitory TMS experiments. Each circle represents each participant. *** and ** indicate $P < 0.001$ and $P < 0.01$ in one-sample $t$-tests ($df$ = 13 for panel **a** and $df$ = 14 for panel **b**), respectively. The error bars show the s.e.m.

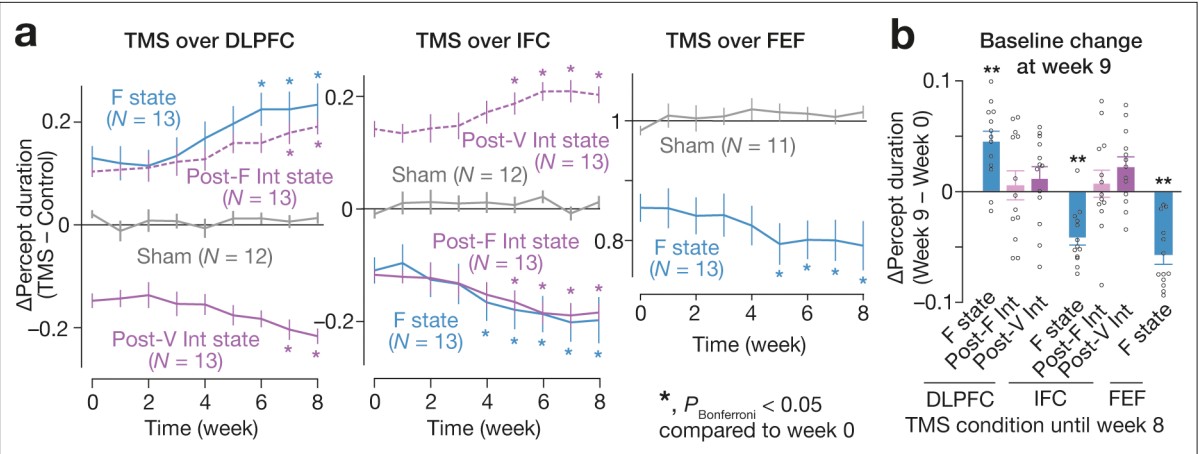

**Figure 10.** Long-term effects. We found the accumulative effects of the state-/state-history-dependent TMS on the bistable visual perception in 2-month longitudinal experiments (**a**). We focused on the seven TMS conditions that induced significant behavioural effects (*Figure 4b,c*). Even after the 2-month weekly TMS experiments, we observed significant changes in the baseline percept duration in three of the seven TMS conditions (**b**). In panel **a**, the Y-axes show the changes in the median percept duration (TMS – Control), and * indicates $P_{\text{Bonferroni}}$ <0.05 in paired *t*-tests with comparisons to the week-0 values (*df* = 12). In panel **b**, the Y-axis indicates the changes in the control performance between week 0 and 9. ** indicates $P_{\text{Bonferroni}}$ <0.01 in one-sample *t*-tests (*df* = 12).

be seen as results of the acceleration of the F-Int-V travel due to the more frequent forward moves from Int to F state, which was induced by the lower Int-to-F energy barrier (*Figure 8g*).

## Replicability

The main behavioural findings were replicated in a small but independent cohort (N = 14; $t_{13}$ >2.9, $P$ < 0.01 in one-sample *t*-tests; *Figure 9*). Moreover, another independent experiment (N = 15) showed that, as in our previous work (*Watanabe et al., 2015*; *Watanabe et al., 2014a*), the excitatory TMS induced behavioural effects opposite to those yielded by the inhibitory stimulation ($t_{14}$ >2.8, $P$ < 0.01 in one-sample *t*-tests; *Figure 9b*), which added indirect but empirical support for the current observations.

## Long-term effects

To examine the potential of future clinical applications of this brain-state-dependent neural stimulation system, we conducted another two-month longitudinal experiment (N = 63) and found accumulative effects of this closed-loop TMS system. Despite its weak power—approximately 78 % of the power of the similar TMS protocol (*Watanabe et al., 2015*; *Watanabe et al., 2014a*)—, the behavioural effects became larger along with the weekly TMS sessions ($t_{12}$ >3.2, $P_{\text{Bonferroni}}$ < 0.05 in paired *t*-tests; *Figure 10a*). Moreover, the F-state-dependent TMS affected the baseline perceptual stability, which was detectable even one week after the 2-month TMS experiments ($F_{6,12}$=16.4, $P < 10^{-3}$ for the main effect in a two-way ANOVA; $P_{\text{Bonferroni}}$ < 0.01 in post-hoc one-sample *t*-tests; *Figure 10b*).

## Discussion

This study has demonstrated that causal roles of the three PFC regions—right DLPFC, IFC and FEF—in spontaneous perceptual inference were behaviourally detectable when we tracked the brain state dynamics underpinning such perceptual fluctuation. The state-/state-history-dependent behavioural causality was explained by the large-scale brain state dynamics and attributable to the energy landscape structures. Moreover, the current findings suggest distinct functions of the PFC regions in terms of the brain state dynamics: the activation of DLPFC enhances the functional integration between the Frontal and Intermediate state, whereas the IFC activity promotes the functional segregation between the two brain states; the FEF activity stabilises Frontal state.

These prefrontal functional differences can be interpreted in more conventional neuropsychological contexts, if we presume that (i) Frontal state is involved in the generation of top-down signals,

(ii) Visual state is related to bottom-up signal generation and (iii) Intermediate state is a place for the interactions of the top-down and bottom-up information. Given these assumptions, the DLPFC activity can be seen as a factor to enhance the generation and flow of top-down information, whereas the IFC activity facilitates those of bottom-up information. The activation of FEF can be regard as a support for the top-down signal generation.

At first glance, this neuropsychological interpretation is well fitted to widely-known concepts on two attention systems in the brain (*Baldauf and Desimone, 2014*; *Corbetta et al., 2008*; *Corbetta and Shulman, 2002*): the dorsal attention system including the FEF is mainly involved in preparing and applying top-down attention, whereas the ventral attention system including the IFC is associated with processing of bottom-up attention signals.

However, as argued in a recent TMS study (*Weilnhammer et al., 2021*), it may be difficult to fully describe the PFC roles during bistable perception from the perspective of such attention systems only. For example, if the main function of the IFC in this experiment is to receive bottom-up attention signals from the visual cortex, distribute them to the fronto-parietal areas and direct the attention to a different perceptual inference, the neural suppression of IFC should prolong the percept duration. By contrast, in this study, the inhibition of IFC activity did not always lengthen the percept duration but often shortened it (*Figure 4b,c*). Given these, it would be more reasonable to infer that the top-down and bottom-up signals, which are supposed to be generated and communicated in the brain state dynamics, contain not only attention-related information but also other types of neural signal such as prediction error in predictive coding paradigm (*Brascamp et al., 2018*; *Hohwy et al., 2008*; *Weilnhammer et al., 2021*; *Weilnhammer et al., 2017*).

By the same logic, it may be difficult to regard the current findings as evidence supporting the notion that the DLPFC and IFC are irrelevant to the bistable perception itself but only involved in reporting the perceptual states (*Brascamp et al., 2015*; *Frässle et al., 2014*). If these prefrontal regions are related to reporting the perceptual awareness, the neural suppression of the PFC areas should always prolong the percept duration. In reality, the inhibitory TMS over PFC could shorten the percept duration (*Figure 4b,c*), which indicates the possibility that the DLPFC and IFC are closely involved in the bistable perception itself. In the meantime, we have to be careful to generalise this indication to other types of bistable perception. In fact, it is not SFM but binocular rivalry that the previous study used to show the negligible associations between the PFC activity and bistable perception (*Brascamp et al., 2015*; *Frässle et al., 2014*). To resolve this situation, future studies would have to examine brain-state-dependent behavioural causality of the PFC in SFM-induced bistable perception using the non-report paradigm (*Tsuchiya et al., 2015*).

In the bistable visual perception paradigm, a line of previous TMS studies reported behavioural causal roles of the parietal cortex in the right hemisphere (*Carmel et al., 2006*; *Kanai et al., 2011*; *Kanai et al., 2010*; *Vernet et al., 2015*; *Zaretskaya et al., 2010*), whereas no investigation found such effects in the prefrontal cortex except for a recent one (*Weilnhammer et al., 2021*). The current findings propose a neurobiological account for this difficulty in detecting the prefrontal causality. As shown in *Figure 4b,c*, the behavioural responses to the TMS depended on the timing of the neural stimulation. Without tracking the brain state dynamics, the behavioural responses to the TMS would be close to the average of such brain-state-dependent effects and likely to be observed as null results (e.g. *Figure 4d*).

If this is the case, why could the recent study successfully detect prolonged percept in the bistable visual perception test after applying conventional TMS over the IFC? (*Weilnhammer et al., 2021*) The TMS protocols are different between the previous study and the current work, and further studies are necessary to answer this question directly; but we can speculate the reason as follows. In the recent study, theta-burst TMS was administered to the IFC during rest. Considering that default-mode network is mainly active during rest and the IFC and DLPFC tend to be inactive, we can speculate that the TMS was applied during brain states similar to Visual state. As shown in *Figure 4b*, such V-state-dependent TMS over the IFC could induce a moderate prolongation of the percept duration (Cohen's $d$ = 0.3), which may be detected as a significant behavioural effect in the recent study.

Methodologically, the current findings can be seen as another work highlighting the fact that certain brain-behaviour causality is changing so dynamically that we cannot behaviourally detect it in conventional neurostimulation methods that do not consider the temporal changes of the brain states (*Bergmann, 2018*; *Karabanov et al., 2016*). Previous work has addressed this issue by controlling,

inferring or monitoring the brain state: some studies controlled the brain state using external stimuli and applied the TMS when a specific sensory stimulus was presented to the participants (*Cattaneo et al., 2010a*; *Cattaneo et al., 2010b*; *Ezzyat et al., 2017*); a clinical research adopted emotional states as indicators of the neural activity and determined the timing of the deep brain stimulation based on such inference (*Scangos et al., 2021*); neurophysiological studies monitored neural activity and determined the timing of brain stimulation based on the frequency, phase or power of the neural signal (*Mrachacz-Kersting et al., 2019*; *Polanía et al., 2018*; *Schaworonkow et al., 2019*; *Stefanou et al., 2018*; *Zrenner et al., 2018*).

The current TMS method is categorised into the last group and could be seen as advancement of such direct-neural-monitoring-based brain stimulation. Differently from the previous work monitoring a single neural activity (*Mrachacz-Kersting et al., 2019*; *Polanía et al., 2018*; *Schaworonkow et al., 2019*; *Stefanou et al., 2018*; *Zrenner et al., 2018*), the TMS system used here can track the brain state using neural activity patterns recorded from multiple remote brain regions. Considering the multi-regional brain state dynamics underpin complex cognitive activities (*Ezaki et al., 2018*; *Watanabe et al., 2014c*; *Watanabe and Rees, 2017*), such a multivariate monitoring approach could be a more effective manner to investigate more physiological brain-behaviour causality.

Moreover, the combination of EEG-triggered TMS and energy landscape analysis may become foundation of a novel tool to control seemingly unstable behaviours. As shown in our longitudinal experiment (*Figure 10*), this brain-state-dependent neural stimulation system has accumulative effects despite its relatively weak stimulation. Given that the perceptual stability seen in this bistable perception test was relevant to autistic cognitive rigidity (*Watanabe et al., 2019a*), this longitudinal observation may become a basis for new clinical non-invasive neural interventions and accelerate the development of such state-dependent brain stimulation methods.

These findings may not be directly applicable to other types of multistable visual perception, such as binocular rivalry, which is linked to lower-level brain architectures such as the visual cortex (*Haynes et al., 2005*; *Lee et al., 2005*; *Leopold and Logothetis, 1996*; *Meng and Tong, 2004*; *Miller et al., 2000*; *Pettigrew and Miller, 1998*; *Polonsky et al., 2000*). In fact, a comprehensive behavioural study reported the relative dissimilarity in perceptual switching rate between the current SFM-induced bistable perception and the binocular rivalry (*Cao et al., 2018*). In contrast, the same study found the similarity in between the SFM-induced bistable perception and other fluctuating perception triggered by spinning dancer (*Liu et al., 2012*) and Lissajous-figure (*Weilnhammer et al., 2014*). Given this, the current observations might be more applicable to types of bistable perception that requires construction of a 3D image from 2D motion compared to the other types such as the binocular rivalry.

To interpret the current observations in psychological contexts such as predictive coding (*Brascamp et al., 2018*; *Hohwy et al., 2008*; *Weilnhammer et al., 2021*; *Weilnhammer et al., 2017*). more model-based neuroimaging studies, attention-tracking behavioural research and theoretical investigations would be necessary.

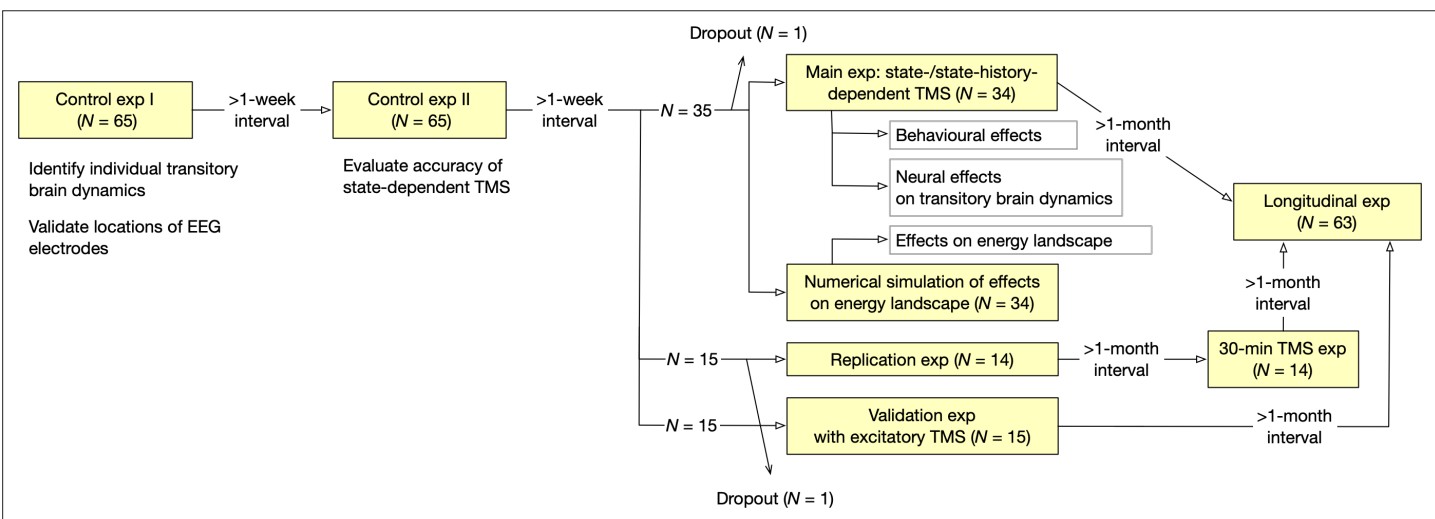

**Figure 11.** Oveall study design. The current study consisted of seven EEG/TMS experiments and one numerical simulation.

This study has resolved the long-lasting controversy over prefrontal causal roles in multistable perception (*Brascamp et al., 2018*) and revealed distinct functions of the PFC in the brain state dynamics that underpin spontaneous perceptual inference. Furthermore, the combination of the brain-state tracking method and neural-activity-dependent brain stimulation system may re-ignite neurobiological investigation on state-dependent dynamic causality (*Silvanto et al., 2008*) in human cognition and become another foundation of more effective neural perturbation tools to intervene in some neuropsychiatric conditions.

## Materials and methods

### Overall study design

This study consisted of seven experiments and one numerical simulation (*Figure 11*).

First, 65 healthy adult individuals underwent two control experiments, in which we recorded their EEG signals while they were experiencing bistable visual perception induced by a structure-from-motion (SFM) stimulus (*Figure 1a*). In the Control Experiment I, The EEG data were used to (i) identify individual brain dynamics during bistable visual perception and (ii) verify the locations of EEG electrodes and TMS stimulation site. In the Control Experiment II, we examined the accuracy of the brain-state tracking and state-dependent TMS system.

Second, 35 of all the 65 individuals participated in the main experiment. In this experiment, using the information about individual brain state dynamics, we administered inhibitory TMS to the participants in state-/state-history-dependent manners. In parallel, we conducted a numerical simulation to predict how inhibitory TMS changed individual energy landscape structures and, resultantly, brain state dynamics. Thirty four of the 35 participants completed this experiment. This sample size of the main experiment was determined on a power analysis (effect size = 0.5; power = 0.8; alpha = 0.05 for paired *t*-tests).

Other 15 participants underwent a replication experiment that re-tested behavioural causality found in the main experiment, and 14 of them completed the experiment. Afterwards, the 14 participants underwent a conventional 30 min quadripulse TMS (QPS) (*Hamada et al., 2007*; *Watanabe et al., 2015*; *Watanabe et al., 2014a*) experiment, in which they received the QPS over one of the three PFC regions (DLPFC, IFC, and FEF) during rest.

The other 15 individuals participated in a validation experiment that examined whether excitatory TMS induced behavioural effects opposite to those caused by inhibitory TMS.

The 63 individuals who completed either of these experiments participated in a longitudinal experiment, which evaluated the accumulative effects of the current neural stimulation method.

### Participants and ethics

All the 65 participants were right-handed adults (Edinburgh Handedness Inventory laterality score = 75 ± 16, mean ± sd). None of them had neurological, psychiatric or other medical history and was free from any contraindication to TMS experiments (*Wassermann, 1998*).

Except for the control experiments and 30 min QPS experiment, all the experiments asked the participants to undergo the tests continually over several weeks; therefore, some of the participants did not complete the study allegedly due to their busyness of life. Consequently, the final sample size was 34 for the main experiment (age = 23.1 ± 1.8; female = 12), 14 for the replication experiment (age = 22.2 ± 2.0; female = 8), 15 for the validation experiment using excitatory TMS (age = 22.7 ± 1.7; female = 7), and 63 for the longitudinal study (age = 22.8 ± 1.9; female = 27). Including the participants who dropped out, no participant reported adverse effects throughout this study.

This study was approved by Institutional Ethics Committees in RIKEN and The University of Tokyo (20-132). The TMS protocols used here complied with the guideline issued by the Japanese Society for Clinical Neurophysiology and that by International Federation of Clinical Neurophysiology (*Rossi et al., 2009*). All the participants provided written informed consents before any experiment and were financially compensated for their participation.

### Device setup: test of bistable visual perception

The test design of bistable visual perception paradigm in this study is essentially the same as that used in our previous work (*Watanabe et al., 2014c*; *Watanabe et al., 2019a*). The participants were

presented with a structure-from-motion (SFM) stimulus (*Figure 1a*), a sphere consisting of 200 sinusoidally moving white dots in a black background (angular velocity, 120°/s) with a fixation cross (0.1° × 0.1°) at the centre of the 27-inch LCD monitor (BenQ PD2710, resolution: 2560 × 1440).

In each run, the participants were instructed to see the SFM stimulus for 90 s with their chins put on a chin rest. They were asked to push one of the three buttons according to their visual perception: one for upward rotation, another for downward rotation, and the other for unsure or mixture perception. After sufficient training sessions, the participants repeated this run six times in all the experiments except for the control one. The stimulus presentation and response recording were conducted with PsychToolbox three in MATLAB (MathWorks, Inc).

The proportion of the mixture perception was sufficiently small in all the participants, all the experiments (1.4% ± 0.5% of all stimulus presentation times, mean ± sd). Thus, we focused on the time during which participants were clearly aware of the direction of the rotation. For each participant, we measured the duration of the clear perception and calculated the median of the duration to evaluate their perceptual stability. The median duration was adopted because the perceptual durations showed long-tailed distributions.

## Device setup: EEG

Throughout entire this study, we recorded EEG signals from seven regions of interest (ROIs) to monitor brain state transitions using a TruScan Research EEG system with 32 TMS compatible Ag/AgCl ring electrodes (Deymed Diagnostic, Czech Republic).

As in our previous work, the seven ROIs consisted of the right FEF (x = 38, y = 0, z = 60 in MNI coordinates), DLPFC (x = 44, y = 50, z = 10), IFC (x = 48, y = 24, z = 9), anterior superior parietal lobule (aSPL; x = 36, y = –45, z = 44), posterior superior parietal lobule (pSPL; x = 38, y = –64, z = 32), lateral occipital complex (LOC; x = 46, y = –78, z = 2) and V5 (hMT/V5; x = 47, y = –72, z = 1) (*Figure 1b*). These coordinates are based on the following previous studies: a study by Sterzer and colleagues (*Sterzer et al., 2002*) for the FEF; one by Knapen and colleagues (*Knapen et al., 2011*) for the DLPFC; one by Kleinschmidt and colleagues (*Kleinschmidt et al., 1998*) for the IFC; three studies by Kanai, Carmel and their colleagues (*Carmel et al., 2010*; *Kanai et al., 2011*; *Kanai et al., 2010*) for the aSPL and pSPL; one by Freeman and colleagues (*Freeman et al., 2012*) for the LOC and V5.

Using a stereoscopic neuro-navigation system (Brainsight Neuronavigation, Rogue Research, UK) and structural MRI brain images, we located the TMS-compatible EEG electrodes right above on these seven ROIs. Also, for the following calculation of Hjorth signals (*Hjorth, 1970*; see Section 3.2.1), we put three other EEG electrodes around each ROI electrode (i.e. four electrodes were used for one ROI; *Figure 12*). The other four electrodes (i.e. 32 electrodes – four electrodes/ROI ×7 ROIs = 4 electrodes) were located on Fpz, Oz, A1, and A2 in accordance with 10–20 international system (*Seeck et al., 2017*). These electrodes were firmly placed on the heads of the participants with elastic caps.

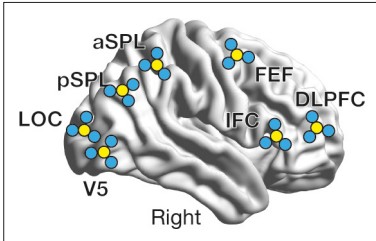

**Figure 12.** Location of electrodes. We placed TMS-compatible EEG electrodes in an original way. First, seven electrodes were located right above on the seven ROIs using a stereoscopic neuro-navigation system. Then, for each ROI, three additional electrodes were placed right around the electrodes for the following Hjorth signal calculation. The yellow circles schematically represent the electrodes put above the ROIs, whereas the blue ones indicate the neighbouring electrodes.

After confirming that the impedance was less than 5 kΩ in all the electrodes, we recorded the EEG signals with a 32-channel amplifier (TruScan EEG LT 32ch Headbox, Deymed Diagnostic; 6 kHz of analogue sampling frequency), in which the signals underwent low-pass filtering (cut-off frequency: 1.25 kHz) and were down-sampled to 3 kHz at almost simultaneously (latency <5 ms).

## Device setup: TMS

To deliver both inhibitory and excitatory stimulation, we used a quadripulse TMS system that can deliver a train of four monophasic magnetic pulses at once from interconnected four magnetic stimulators (70mm-diameter double coil, DuoMag MP, DeyMed Diagnostic, Czech Republic).

To limit the neural effects of TMS into the target site, our EEG-triggered TMS system did

not use conventional quadripulse TMS protocols (*Hamada et al., 2009*; *Hamada et al., 2007*; *Seeck et al., 2017*; *Shirota et al., 2010*; *Watanabe et al., 2014a*) in which the TMS was continually administered throughout 30 min and whose after-effects are observable in connections with remote brain areas (*Watanabe et al., 2015*; *Watanabe et al., 2014a*) and known to last for more than 60 min. Instead, in the main experiment, we conducted a single set of four-pulse 20 Hz monophasic TMS for neural suppression, whereas a set of four-pulse 200 Hz monophasic TMS was administered as an excitatory stimulation in the validation study.

In addition, to obtain sufficient length of clean EEG data for the following brain-state tracking, we put ≥9 sec intervals between the stimulations and set the intensity of the TMS stimulation at a lower level (70 % of active motor threshold, AMT) compared to that in the conventional quadripulse TMS protocol (90 % of AMT) (*Watanabe et al., 2015*; *Watanabe et al., 2014a*). In EEG-triggered TMS, such weak stimulations were considered to induce sufficiently large neural effects (*Schaworonkow et al., 2019*).

The AMT was calculated based on motor evoked potentials (MEPs) of the right first dorsal interosseous (FDI) muscle of each participant in prior single-pulse TMS experiments: the AMT was set as the lowest TMS intensity that evoked a small response ( > 100μV) when the participants maintained a slight contraction of the right FDI (approximately 10 % of the maximum voluntary contraction) in more than the half of 10 consecutive trials (*Watanabe et al., 2015*; *Watanabe et al., 2014a*).

The MEPs of the right FDI were recorded using a pair of 9mm-diameter Ag/AgCl surface cup electrodes that were placed over the muscle belly and the metacarpophalangeal joint of the index finger (*Hanajima et al., 2001*). Before offline analyses, the MEP signals underwent a temporal filter (100 Hz–3 kHz). The optimal place for the single-pulse TMS for the right FDI muscle was determined as the area over which the simulation induced the largest MEP.

The mean AMT across the entire participant cohort in the three types of the current TMS experiment was 35.8% ± 7.9% of the maximum stimulator output.

In one TMS condition, we placed the TMS coil over one of the PFC areas in the same method using the stereoscopic neuro-navigation system (Brainsight Neuronavigation, Rogue Research, UK). We confirmed that the coil did not substantially move throughout the experiment by re-measing the location with the neuro-navigation system at the end of each experiment day. The green circles in *Figure 4a* show such end-of-the-day locations averaged across the four-day sessions in the main experiment.

## EEG analysis

In this study, we analysed the EEG data in both offline and online manners. We described the details of the two types of EEG analysis in the following sections.

## Offline EEG analysis: preprocessing

The offline EEG analysis was applied to data obtained in the control experiment. We conducted the following conventional preprocessing (*Watanabe et al., 2019b*) to the EEG data using MATLAB (MathWorks, US) and EEGLAB (*Delorme and Makeig, 2004*). Note that we used no behavioural response, such as the timing of perceptual switch, in the following preprocessing procedure and energy landscape analysis.

First, the EEG data were referenced to the average across all the electrodes, down-sampled to 300 Hz and underwent a temporal filter (1–80 Hz). Then, we conducted an independent component analysis (ICA) to remove cardio-ballistic artefacts and other artefacts induced by eye blinks, eye movements and muscle activity. The ICA algorithm was based on short-time Fourier transforms and complex-valued version of FastICA with a robust measure of non-Gaussianity (*Hyvärinen et al., 2010*).

Next, we marked epochs whose mean global field power was too large ( > 5 SD of mean power across entire recording) and excluded those time periods in all the following main analysis. We then filtered the data to delta (1–4 Hz), theta (4–8 Hz), alpha (8–13 Hz), beta (13–30 Hz), and gamma (30–80 Hz) bands and estimated a Hilbert envelope amplitude for the gamma-band signal (*Deligianni et al., 2014*; *Watanabe et al., 2019b*).

We used the Hilbert envelope amplitude for the gamma band as a neural signal for each electrode, because the aim of the current EEG recording was to trace the brain state dynamics that was seen in our previous fMRI study (*Watanabe et al., 2014c*) and the gamma-band signal dynamics were

correlated with fMRI signals (*Deligianni et al., 2014*; *Watanabe et al., 2019b*). Finally, after removing autocorrelation, we calculated a Hjorth signal for each ROI in the following sum-of-difference manner (*Hjorth, 1970*): Hjorth signal for ROI$_i$ = (electrode just above ROI$_i$ – surrounding electrode 1) + (electrode just above ROI$_i$ – surrounding electrode 2) + (electrode just above ROI$_i$ – surrounding electrode 1).

## Offline EEG analysis: fitting of pairwise maximum entropy model

We then conducted the energy landscape analysis (*Ezaki et al., 2017*; *Gu et al., 2018*; *Kang et al., 2017*; *Watanabe et al., 2014c*; *Watanabe and Rees, 2017*) of the preprocessed datasets of the seven ROIs. For each participant, the EEG signals were concatenated across different runs. Then, as in our previous fMRI work on the brain dynamics (*Ezaki et al., 2017*; *Watanabe et al., 2014c*; *Watanabe et al., 2014b*; *Watanabe and Rees, 2017*), we binarised each ROI time-series data using the temporal average of the signals as the thresholds. After this operation, a neural activity pattern of the seven ROIs at time $t$ was described such as $V^t = [\sigma_1^t, \sigma_2^t, \ldots, \sigma_N^t]$, where $\sigma_i^t$ represents a binary activity of ROI$_i$ at time $t$ (i.e. $\sigma_i^t = +1 \vee -1$) and $N$ denotes the number of the ROIs (here, N = 7).

## Fitting a pairwise maximum entropy model

The first goal of the energy landscape analysis is to calculate a hypothetical energy value for each neural activity pattern $V_k \left(1 \leq k \leq 2^N, N = 7\right)$. The energy value is not related to metabolic consumption but an index that shows the stability of each state: the more stable and more frequent neural activity pattern should have the lower energy value.

To this end, we fitted a pairwise maximum entropy model (MEM) to the seven binary time-series signals (*Watanabe et al., 2014c*; *Watanabe and Rees, 2017*). We adopted this model because of its simplicity. It consists of two types of parameters: $h_i$ and $J_{ij}$. The $h_i$ represents the basal activity of ROI$_i$ and $J_{ij}$ indicates a pairwise interaction between ROI$_i$ and ROI$_j$. In terms of neurobiology, this model simply assumes that different brain regions have different intrinsic activity $h_i$ and different pairs of brain areas have different coupling strengths $J_{ij}$.

Now, how do we determine the two parameters? In the energy landscape analysis, we determine $h_i$ and $J_{ij}$ so that the model-based ROI average activity $\langle\sigma_i\rangle_m$ and model-based average pairwise interactions $\langle\sigma_i\sigma_j\rangle_m$ is sufficiently close to the average ROI activity $\langle\sigma_i\rangle$ and average pairwise interaction $\langle\sigma_i\sigma_j\rangle$. We calculated the model-based mean ROI activity $\langle\sigma_i\rangle_m = \Sigma_{l=1}^{2^N}\sigma_i\left(V_l\right)P\left(V_l\right)$ and model-based mean pairwise interaction $\langle\sigma_i\sigma_j\rangle_m = \Sigma_{l=1}^{2^N}\sigma_i\left(V_l\right)\sigma_j\left(V_l\right)P\left(V_l\right)$, where $\sigma_i\left(V_k\right)$ is the binary activity of ROI$_i$ in the activity pattern $V_k$ and $P\left(V_k\right)$ is the appearance probability of an neural activity pattern $V_k$.

This appearance probability is given as $P\left(V_k\right) = e^{-E\left(V_k\right)}/\Sigma_{l=1}^{2^N}e^{-E\left(V_l\right)}$, where $E\left(V_k\right) = -\Sigma_{i=1}^{N}h_i\sigma_i\left(V_k\right) - (1/2)\Sigma_{i=1}^{N}\Sigma_{j=1}^{N}J_{ij}\sigma_i\left(V_k\right)\sigma_j\left(V_k\right)$. This formulation of $P\left(V_k\right)$ is determined by the principle of maximum entropy. That is, the information entropy of $P\left(V_k\right)$ is maximised by making the $P\left(V_k\right)$ obey Boltzmann distribution. In other words, to maximise the information entropy of $P\left(V_k\right)$ and minimise any possible constraints, we set the $P\left(V_k\right)$ in the form of Boltzmann distribution.

Based on this definition, we adjusted $h_i$ and $J_{ij}$ until these the $\langle\sigma_i\rangle_m$ and $\langle\sigma_i\sigma_j\rangle_m$ were approximately equal to the empirically obtained $\langle\sigma_i\rangle$ and $\langle\sigma_i\sigma_j\rangle$ using a gradient ascent algorithm.

The accuracy of this MEM fitting was examined by estimating a Pearson correlation coefficient between the model-based appearance probability and empirically obtained appearance probability and calculating a proportion of Kullback-Leibler (KL) divergence in this second-order model ($D_2$) to that in the first-order model ($D_1$) as follows *Watanabe et al., 2014c*; *Watanabe et al., 2013*; *Watanabe and Rees, 2017*: ($D_1 - D_2$)/$D_1$. In the control experiment, the Pearson correlation was larger than 0.95 and the KL-divergence-based accuracy was larger than 84 % (*Figure 2a*).

## Offline EEG analysis: disconnectivity graph in energy landscape analysis

Next, we built an energy landscape and searched for major brain states. The energy landscape was defined as a network of brain activity patterns $V_k$ (k = 1, 2, …, 2 N) with their energy $E(V_k)$, in which two activity patterns were regarded as adjacent if and only if they took the opposite binary activity at a single ROI. We then searched for local energy minima, whose energy values were smaller than those of all the $N$ adjacent patterns.

We then examined hierarchal structures between the local minima by building disconnectivity graphs as follows *Watanabe et al., 2014c*; *Watanabe and Rees, 2017*: (i) first, we prepared a so-called hypercube graph, in which each node representing a brain activity pattern was adjacent to the N neighbouring nodes. (ii) Next, we set a threshold energy level, $E_{threshold}$, at the largest energy value among the 2 N nodes. (iii) We then removed the nodes whose energy values were≥ $E_{threshold}$. (iv) We examined whether each pair of local minima was connected by a path in the reduced network. (v) We repeated steps (iii) and (iv) after moving $E_{threshold}$ down to the next largest energy value. We ended up with a reduced network in which each local min was isolated. (vi) Based on the obtained results, we built a hierarchical tree whose leaves (i.e., terminal nodes down in the tree) represented the local minima and internal nodes indicated the branching points of different local minima.

## Offline EEG analysis: structure of energy landscape

Based on this dysconnectivity graph, we then estimated basin sizes of the local minima as follows. We first chose a node *i* from the 2 N nodes. If any of its neighbour nodes had a smaller energy value than the node *i*, we moved to the neighbour node with the smallest energy value. Otherwise, we did not move, indicating that the node was a local min. We repeated this protocol until we reached a local min. The initial node *i* was then assigned to the basin of the local min that was finally reached. This classification procedure was repeated for all the 2 N nodes. The basin size was defined as the fraction of the number of the nodes belonging to the basin.

The energy barrier between local minima l and *m* was defined based on the procedure of building the disconnectivity graph. When we built the disconnectivity graph by lowering the threshold energy level $E_{threshold}$, we searched for the lowest $E_{threshold}$ at which the two local minima were still connected. The height of the energy barrier from local min l to *m* was then defined as the difference between the $E_{threshold}$ and the energy value for the local min l, whereas that from local min *m* to l was defined as the difference between the $E_{threshold}$ and the energy value for the local min *m*.

Then, as in our previous work (*Watanabe et al., 2014c*), we summarised the local minima as follows: if the energy barriers from the local min l to *m* was lower than a threshold ( = 1, here) and the energy value of the local min l was larger than that of local min *m*, we regarded the local min l and its basin as elements of the basin of the local min *m*. By repeating this procedure, we found that in all the participants their brain activity patterns during bistable visual perception could be classified into any of the three major basins, which corresponded to Frontal-area-dominantly-active, Visual-area-dominantly-active and Intermediate state (F, V and Int state). The energy barrier threshold for this summarisation was set at the same value as in our previous work (*Watanabe et al., 2014c*).

Through this coarse-graining procedure, we defined the three major brain states and calculated structural indices of the energy landscapes (i.e. the height of the energy barrier). In addition, this summarisation allowed us to classify all the nodes (i.e. brain activity patterns) on the energy landscape—except for nodes on the saddles—into any of the three major brain states for each participant. The vectors that were not classified into any of the three major brain states were labelled as 'other state'. This classification information would be used in the following EEG-triggered state-dependent TMS experiment.

Note that a 'brain state' in this study is not a so-called 'miscrostate' in conventional EEG research; it indicates an activity pattern of multiple (here, seven) brain regions or a group of such activity patterns. Although other analyses, such as hidden Markov model (HMM), can also identify brain states (*Baker et al., 2014*; *Ezaki et al., 2021*; *Miller and Katz, 2010*; *Vidaurre et al., 2018*), we adopted the energy landscape analysis in this study because it was previously used to identify the brain states underpinning the bistable visual perception (*Watanabe et al., 2014c*).

## Offline EEG analysis: simulation of brain state dynamics

In the final part of the energy landscape analysis, we probed the brain state dynamics by a random-walk simulation on the energy landscape (*Watanabe et al., 2014c*; *Watanabe and Rees, 2017*). This simulation was performed based on a Markov chain Monte Carlo method with the Metropolis-Hastings algorithm (*Girvan and Newman, 2002*; *Massen and Doye, 2005*).

In this simulation, any brain activity pattern $V_i$ could move only to a neighbouring pattern $V_j$. In other words, this simulation allowed the brain activity pattern to change the activity of only one ROI. Technically, first, one of such neighbouring patterns was randomly chosen. Then, whether actual

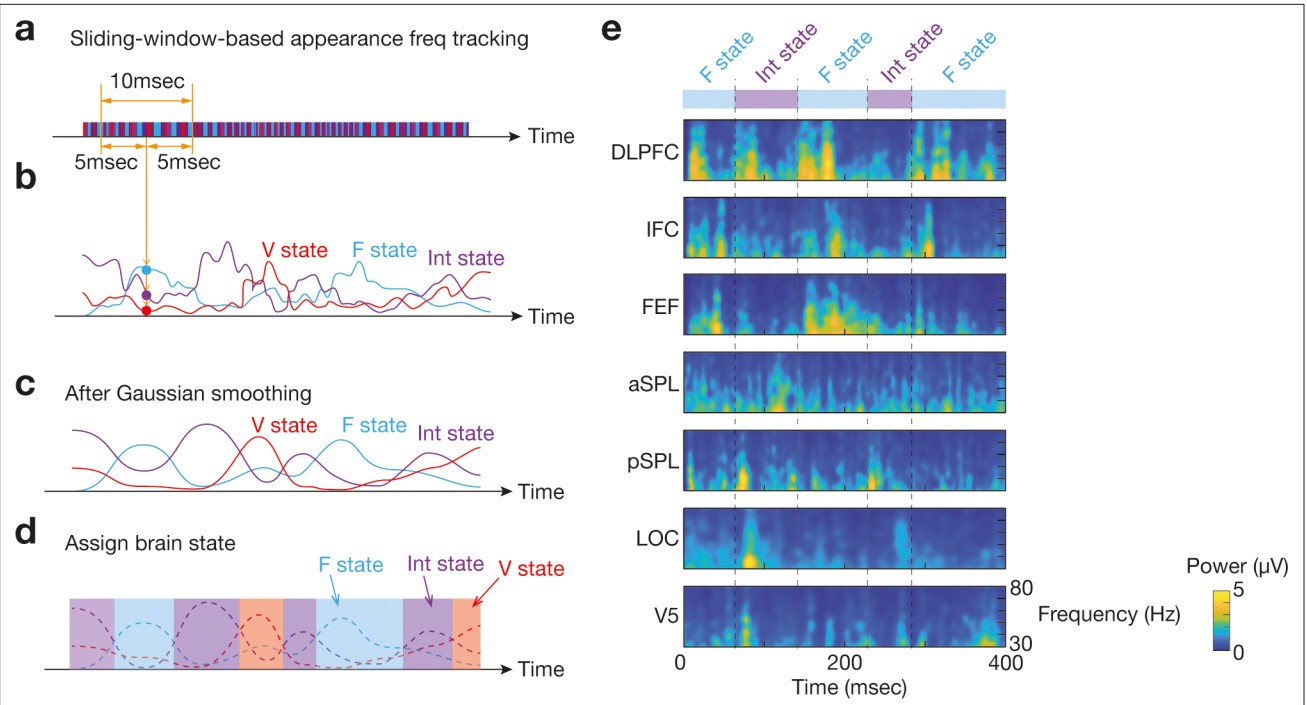

**Figure 13.** Identification of brain state dynamics. To reduce the effects of noise-oriented fluctuation in bran state dynamics, we applied two temporal smoothing procedures to the original brain state dynamics data. First, in a sliding window manner, we calculated three appearance frequency values for the three brain states (**a** and **b**). Then, Gaussian temporal smoothing was applied to the three appearance frequency curves (**c**). By comparing the smoothed curves, we identified which brain state was the most dominant at each timepoint (**d**). Panel **e** shows an example for such brain state transitions with time-frequency plots of the seven ROIs in Participant 1.

movement occurred or not was determined at the probability $P_{ij} = min\left[1, e^{E(V_i) - E(V_j)}\right]$. That is, if the $V_i$ was more unstable than $V_j$ (i.e. $E(V_i) > E(V_j)$), the brain activity pattern should always move from $V_i$ to $V_j$. This rule enhanced the movement to local minima. In the meantime, even if the $V_i$ was more stable than $V_j$ (i.e., $E(V_i) < E(V_j)$), there was some room to move to $V_j$, which prevented the brain activity pattern from being trapped in a local minimum forever.

For each individual, we repeated this random walk $10^5$ steps with a randomly chosen initial pattern and obtained a trajectory of the brain activity pattern such as $\left[V^1, V^2, \dots, V^{10^5}\right]$. After discarding the first 100 steps to minimise the influence of the initial condition, we then classified all the $V^t$ into either of the major brain states (i.e. Frontal, Visual, and Intermediate state) and converted $\left[V^{101}, V^{102}, \dots, V^{10^5}\right]$ to, for example, [Frontal, Frontal, Intermediate, …., Visual]. Finally, we counted how long each of the major brain states continued in the brain state trajectory (dwelling time) and how often one major brain state transited to another major state (transition frequency).

Our previous work demonstrated a strong correlation between the percept duration and the length of return travel between Frontal state and Visual state via Intermediate state (*Watanabe et al., 2014c*). Given this, we compared the behaviourally observed percept duration to the length of the F–Int–V–Int–F travel that was calculated in the above random-walk simulation.

## Offline EEG analysis: temporal smoothing

In parallel with the random-walk simulation, we examined empirical brain state dynamics probing the binary neural vectors with seven elements, $V^t$. For this purpose, we first categorised all $V^t$ into either of the three major brain states based on the classification information that was obtained in the above 'Offline EEG analysis: structure of energy landscape' section. The vectors that were not classified into any of the three major brain states were labelled as 'Other state'.

To reduce the effects of signal fluctuation, we then applied the following temporal smoothing to the time-series of the brain states. First, in a sliding window manner (window length = 10 ms), we

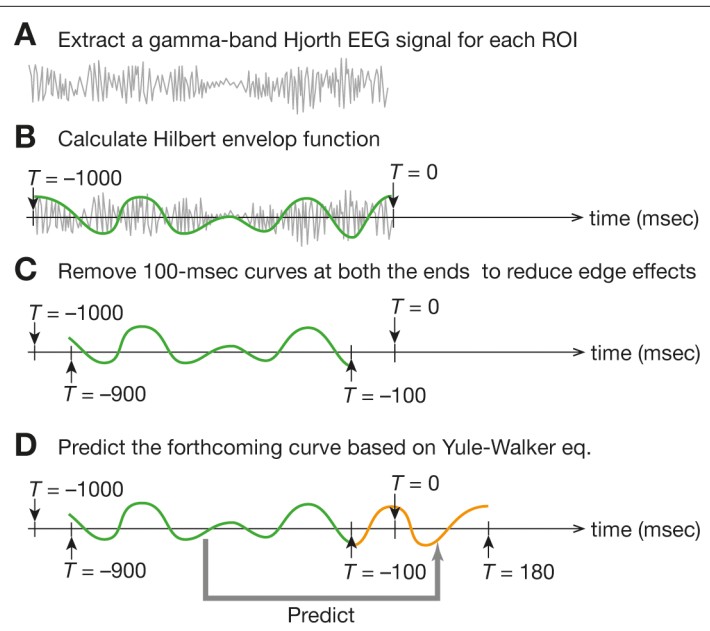

**Figure 14.** Online EEG preprocessing . For the online preprocessing of EEG data, we first calculated a Hjorth EEG signal from each ROI data and then extracted a gamma-band signal in a sliding-window manner (**a**). Next, we fitted the Hilbert envelop function (**b**) and removed both ends of the 1000 ms time window to reduce the edge effects (**c**). Finally, based on the remaining 800 ms Hilbert envelop function, we made the prediction of the forthcoming 280 ms signal (**d**). Such 280 ms signals would be used for brain-state tracking.

calculated the appearance frequency for each brain state in the time window (*Figure 13a*), which was assigned to each centre time point in the window as the representative appearance frequency for each brain state (*Figure 13b*). Next, a Gaussian smoothing filter (FWHM = 10 ms) was applied to the representative appearance frequency curves (*Figure 13c*). Based on the resultant appearance frequency values, we chose the most frequent brain state and assigned it as the brain state at the time point (*Figure 13d and e*). Note that this temporal smoothing eliminated the 'other state'.

This empirical brain state dynamics would be used to evaluate the accuracy of the following online EEG analysis.

## Online EEG analysis: preprocessing

The Online EEG analysis was conducted in all the state-dependent TMS experiments, and its protocol was conceptually the same as that in the previous studies (*Chen et al., 2013*; *Schaworonkow et al., 2019*; *Stefanou et al., 2018*; *Zrenner et al., 2018*). The preprocessed EEG signals ( < 1.25 kHz and 3kHz-downsampled) were input to the real-time target PC machine through a DAQ board (sampling rate, 2 kHz), in which Simulink Real-Time model and xPC Target in Simulink (MathWorks, US) were running to analyse the EEG signals and trigger the TMS system via a TTL signal. We set the analysis model in the target PC through an Ethernet-connected host PC prior to the experiments for each participant.

First, using the 'sum-of-difference' manner, we converted the EEG signals from 28 electrodes (four electrodes/ROI × 7 ROIs) into seven Hjorth signals at each timepoint. Then, in a sliding-window manner (window length = 1000 ms, i.e. 2000 time points), we extracted the gamma-band signals (30–80 Hz) by applying a fast Fourier transformation (FFT) to EEG data in every time window (*Figure 14a*). The gamma band was chosen because (i) this EEG analysis aimed at reproducing our previous fMRI findings (*Watanabe et al., 2014c*) and (ii) the EEG signals in the frequency window is considered to have temporal properties similar to those of fMRI signals (*Deligianni et al., 2014*; *Watanabe et al., 2019b*). We then estimated the Hilbert envelope function for the gamma-band signals, whose amplitude would be used as neural signals (*Watanabe et al., 2019b*; *Figure 14b*). To reduce the edge effects (*Stefanou et al., 2018*; *Zrenner et al., 2018*), we trimmed 100 ms of the neural time-series data on both the edges of the time window (*Figure 14c*).

Using the remaining 800 ms of the neural signals, we then conducted an autoregressive forward prediction based on Yule-Walker equation (order = 30)(*Chen et al., 2013*; *Stefanou et al., 2018*; *Zrenner et al., 2018*) and estimated 280 ms of neural signals that would come after the edge of the trimmed data (*Figure 14d*). That is, given the 100 ms edge trimming, this calculation predicted the neural signal in a period between $T = -100$ ms and $T = 180$ ms when we set $T = 0$ at the actual recording timing (i.e. the original terminal edge of the EEG data). The length of the forward prediction was set at 180 ms because (i) one inhibitory TMS in this study took 150 ms, (ii) the temporal smoothing required 5 ms more data points for its sliding-window-based calculations (see Section Offline EEG analysis: temporal smoothin), and (iii) we prepared 25 ms buffer for the following signal processing to realise a nearly simultaneous EEG-triggered TMS system (*Chen et al., 2013*).

We obtained such 280 msec time-series signal for each ROI and used them in the following analysis.

## Online EEG analysis: binarisation

We then conducted online binarisation of the neural signals during the predicted time period. The binarisation threshold was calculated for each ROI based on the EEG data obtained in a control run that was conducted right before every TMS experiment. Technically, we applied the above-stated "Online EEG analysis: preprocessing" procedure to the EEG data during the control run and calculated the average of the Hilbert envelop amplitude for each ROI in each participant.

## Online EEG analysis: brain-state-/history-dependent TMS

These preprocessing and binarisation processes yielded a binary neural vector with seven elements ( + 1 or –1) at each time point in the 280 ms period (from $T = -100$ to $T = 180$). We then categorised the binary neural vectors into either of the three major brain states based on the classification information that was obtained in the offline energy landscape analysis in the control experiment. The binary neural vectors that were not categorised into any of the three major brain states were labelled as 'Other state'.

Next, to reduce the effects of signal fluctuation, we applied the same temporal smoothing protocol to the resultant brain state vector as that used in the 'Offline EEG analysis: temporal smoothing'. As a

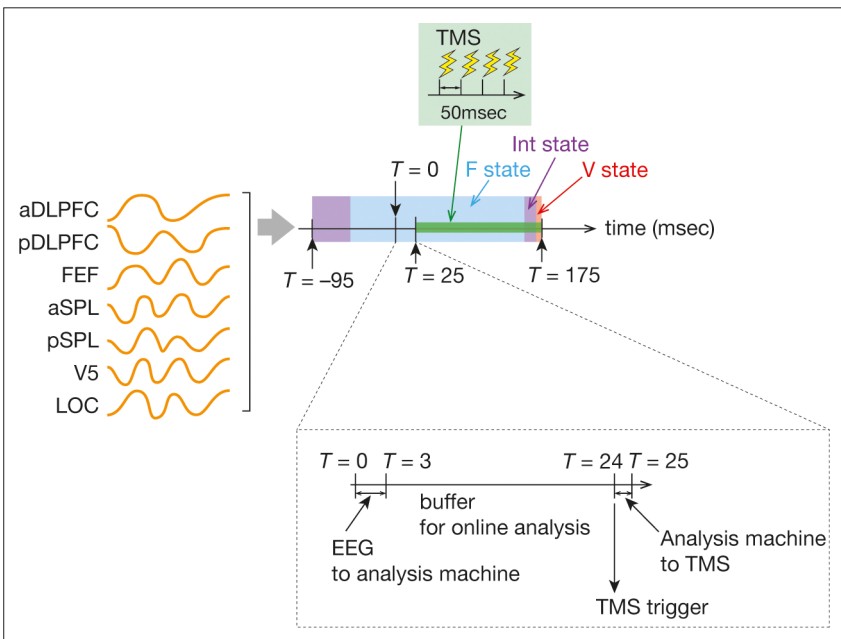

**Figure 15.** Timings of TMS. After the online preprocessing, we obtained seven time-series data for the seven ROIs (orange curves in the left). We put the seven ROI signals into binarisation, brain state categorisation and temporal smoothing procedures, which resulted in brain state dynamics in the 270 ms time period ($T = -95$ to $T = 175$ ms). When one brain state dominated the time window, the online analysis machine sent a trigger at $T = 24$ ms after a 21 ms buffer period (a dashed box in the bottom). A burst of the inhibitory TMS consisted of three TMS stimulations with 500 msec intervals and thus took 150 ms (green box in the centre).

result of these analyses and smoothing procedures, we obtained a series of representative brain states in a period between $T = -95$ ms and $T = 175$ ms. Note that this temporal smoothing eliminated the 'Other state'.

The state-dependent TMS was performed based on the brain states that were predicted to appear in the forthcoming period between $T = 25$ ms and $T = 175$ ms. The TMS was administered only when the target brain state was dominant in the period ( > 90% of the brain states). For example, for the Frontal-state-dependent TMS, only if Frontal state was predicted to dominantly appear in more than 90 % of the time period, the real-time analysis machine sent a TTL trigger signal to the TMS device 21 ms after the EEG signals were input to the analysis machine.

This 21 msec buffer for the online EEG analysis was chosen because the EEG data took ~3 ms to reach the analysis machine and the TTL signal from the analysis machine took ~1 msec to trigger the TMS. Given these latencies, the TMS was supposed to start almost 25 ms after the EEG recording ($T = 25$ ms; *Figure 15*).

In addition, the 21 ms buffer was sufficiently long for the online EEG analysis. By connecting the TTL signal back to the DAQ board of the real-time analysis machine, we evaluated the signal processing delay using the EEG data collected in the Control experiment I and confirmed that the TTL signal reached back to the analysis machine 21.3 ± 0.02 ms (mean ± s.d., 21.1 ms–21.5 ms) after the EEG signals were input into the machine.

For the neural-history-specific TMS, we looked into both the forthcoming period ($T = 25$ ms to $T = 175$ ms) and the preceding period ($T = -95$ ms to $T = 25$ ms). The TMS was administered only when both the two periods dominantly showed the target brain states. That is, for example, Post-Frontal Intermediate-specific TMS was administered only when Intermediate state was dominant in the forthcoming period ( >90% of the brain states) and Frontal state was dominant in the neighbouring time window.

Note that, for both the brain-state-/history-specific stimulation, once a TMS was administered, we did not conduct the next stimulation until at least 9 s passed. Such an interval gave the current brain-state-dependent TMS system a sufficient length of clean EEG data before the next TMS.

## Experiment design and statistics

In the above sections, we stated all the essential device setups and analysis procedures. In the following sections, we elaborated on the actual designs of the experiments using such devices and analyses protocols.

## Control experiments

The control experiment consisted of two parts: EEG part and EEG/TMS part. The Control Experiment I was conducted to (i) identify the brain state dynamics in the offline EEG analysis and (ii) validate the locations of the EEG electrodes, whereas Control Experiment II was performed to (iii) verify the accuracy of the online EEG analysis and brain-state-dependent TMS system. Both the parts employed the same 65 individuals and were performed at least two-day intervals.

## Control experiment I: EEG part (day 1)

In the Control Experiment I, we collected EEG signals while the participants were conducting the test of bistable visual perception (1.5 min/run × 10 runs). The participants started this EEG recording sessions after sufficient training of the test.

For the aim (i), we applied the offline energy landscape analysis to the EEG data and examined whether the brain dynamics seen in our previous fMRI study (*Watanabe et al., 2014c*) were qualitatively reproduced in the current EEG experiment. This analysis was conducted for the aim (ii) as well: if we successfully confirmed the reproducibility, such observations would provide face validation to the locations of the EEG electrodes. The details of this offline energy landscape were stated above (see 'Offline EEG analysis' sections).

## Control experiment II: EEG/TMS part (Day 2)

In the Control Experiment II, the participants underwent test of bistable visual perception (1.5 min/run × 11 runs) with the brain-state-dependent TMS system, which was almost the same as stated in the sections (see 'Device setup: TMS' and 'Online EEG analysis') except for the locations of the TMS

coil and an EEG electrode. We placed one of the 32 TMS-compatible Ag/AgCl ring electrodes—which was located on A1 in the original setting—on a wooden table that was set remotely from the participants. The TMS coil was placed over the electrode. Using the signal from this electrode, we measured when each TMS stimulation was conducted without causing significant artefacts on the EEG data.

The scalp EEG data in the first run were used to calculate the threshold value for binarisation. Therefore, we did not apply TMS in the run. In the rest ten runs, the TMS was performed in the following five different conditions: Frontal-/Intermediate-/Visual-state-dependent and Post-Frontal-/Post-Visual-Intermediate-state-dependent conditions. Each temporal condition was tested in two runs.

In the analysis, we performed both the offline and online analyses using the same EEG data. As stated above, the online EEG analysis required (i) the classification information to determine which major brain state would be assigned to each neural activity pattern and (ii) the threshold to binarise the neural signals. The requirement (i) was met by employing the results of the first half of the control experiment. For the requirement (ii), the data during in the first run were used: we conducted the preprocessing part of the online EEG analysis, extracted a Hilbert envelop curve and calculated the mean value of the envelope amplitude for each ROI in each participant.

Using the classification information and binarisation threshold, we performed the online analysis with the EEG data recorded during the rest of the runs (10 runs). This online EEG analysis was the same as that described above 'Online EEG analysis' sections, and we obtained a time-series vector representing brain state dynamics.

For the sake of comparison, the offline analysis also used the EEG data recorded in the 2nd-11th run and estimated the brain-state dynamics. We applied all the above-mentioned 'Offline EEG analysis' procedures to the data except for the last random-walk simulation part. This offline analysis provided us with information about which major brain state should be assigned to each neural activity pattern. Based on the classification information, we then labelled the preprocessed EEG time-series data, which resulted in a time-series vector of brain state dynamics.

In sum, these two types of EEG analysis gave us two time-series vectors representing the brain state dynamics for each participant. We then compared the two vectors and estimated how accurately the online analysis-based time-series vector predicted one that was based on the offline analysis. Technically, we counted timepoints whose brain states in the online analysis-based vector were the same as those in the offline analysis-based vector. We repeated such counting for each of the three major brain states in each participant and evaluated the accuracy.

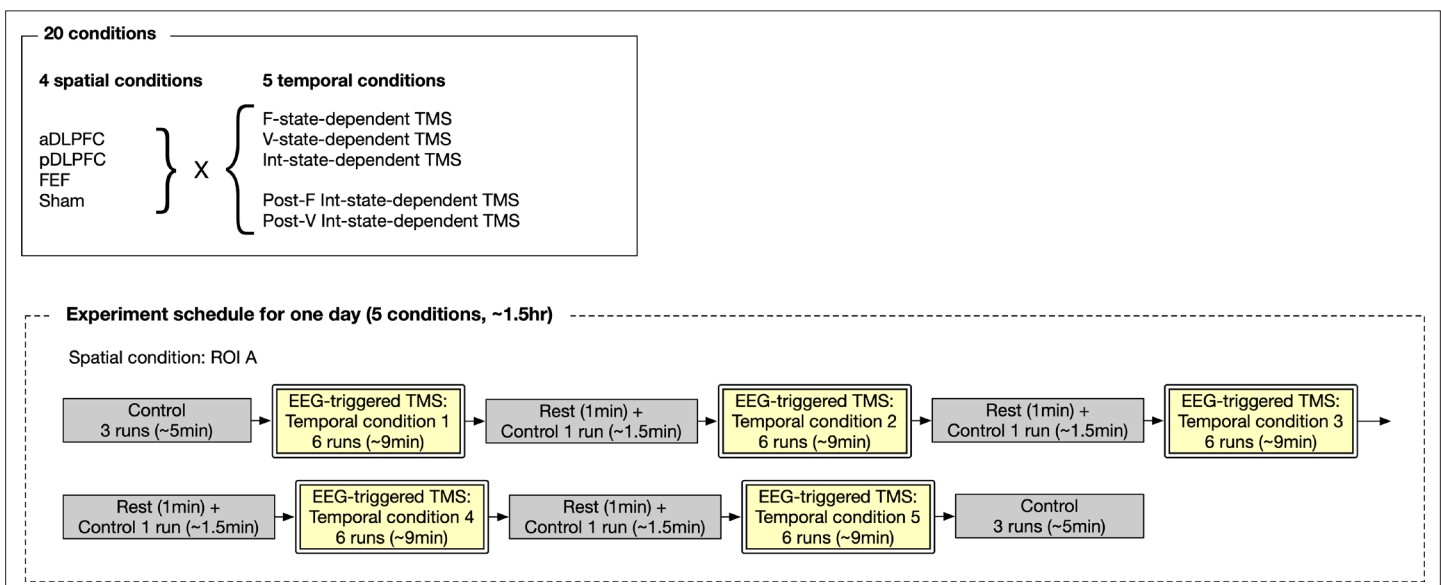

**Figure 16.** Design of the main experiment. For each participant, the experiment was conducted over four different days with> one week intervals. On each day, they received TMS/sham over one of the three PFC days in five different timings. The stimulation site for the sham condition was randomly chosen from the three PFC areas and balanced across the participants.

In parallel, we examined the temporal accuracy of the TMS. The brain-state-dependent TMS system was designed to administer a TMS train 25 ms after a specific brain state was detected in the EEG signals. Using the signal from the EEG electrode that was placed on the wooden table and covered by the TMS coil, we measured the actual latency from the brain-state detection to the TMS stimulation and calculated the difference between the empirical latency and the presumed time-lag (i.e. 25 ms). We repeated this estimation for all the five temporal conditions in each participant.

## Main experiment: design

In the main experiment, we examined causal behavioural effects of state-/state-history-dependent TMS on the median percept duration in a test of SFM-induced bistable visual perception. The participants underwent TMS over three different brain sites (DLPFC, IFC and FEF) in five different neural timings (F-. V-, Int-, Post-F Int- and Post-V Int-state-dependent TMS). In addition, we conducted an experiment using sham stimulation, in which mock TMS was applied over one of the three PFC sites. The choice of the target site in the sham condition was randomised and balanced across the participants.

These experiments were conducted on four different days with more than one-week intervals. In each day, the participants received TMS over the same brain sites in the five different brain-state conditions (*Figure 16*): they underwent six runs of bistable visual perception tests for each condition, between which they had one-min rest and one control run without TMS. The order of the brain-state conditions was randomised and balanced across the participants. Before and after the TMS conditions, the participants also performed longer control runs.

## Main experiment: behavioural and neural analysis

Behaviourally, we calculated the median percept duration for each TMS condition and control session. The behavioural effects of the TMS were evaluated as the ratio of the median percept duration in the TMS conditions to that in the control runs conducted at the beginning of the entire main experiment.

Neurobiologically, we examined TMS's effects on the brain state dynamics. Using the results of the online EEG analysis, we evaluated the dwelling time lengths for the three major brain states and transition frequencies between them after each TMS administration. To reduce the artefacts induced by the TMS, we discarded EEG signals recorded between the beginning of the TMS ($T$ = 25 ms) and 100 ms after the end of the TMS ($T$ = 175 ms + 100 ms = 275 ms) and analysed the data in the first 4 s time window in the remaining data (i.e. the period from $T$ = 275 ms to $T$ = 4275 ms).

## Main experiment: numerical simulation of TMS effects on energy landscape

To investigate the neurobiological changes in the brain state dynamics, we also numerically examined how the inhibitory TMS affected the structures of the individual energy landscape, such as the height of the energy barrier.

This simulation was conducted based on the $H_i$ and $J_{ij}$ that were obtained in the offline energy landscape analysis using the EEG data recorded during the control experiment. If ROI$_i$ was the target site of the inhibitory TMS, we removed all the binary neural vectors $V^t$ whose element $i$, $\sigma_i$ , was set at +1 (i.e. active). Then, using the same $H_i$ and $J_{ij}$ , we built the disconnectivity graph (see 'Offline EEG analysis: disconnectivity graph in energy landscape analysis') and estimated the structural properties of the energy landscape (see 'Offline EEG analysis: structure of energy landscape'). When the local minimum that represented any of the major brain states was removed in this simulation, we alternatively used the brain state that included the neighbouring neural vector as the major brain state. The neighbouring neural vector was defined as a vector that was different from the vector for the local minimum only in one element. In such numerical simulation, all the three major brain states were preserved whichever PFC site was inhibited.

## Main experiment: comparison between the numerical simulation, neural effects and behavioural causality

We compared the results of the numerical simulation, changes in the brain state dynamics and causal behavioural effects. First, we built hypotheses about the changes in the brain state dynamics based on the numerical simulation and tested them by comparing the results of the numerical simulation with

the effects on the brain state dynamics. After confirming the validity of the hypotheses, we examined the relationship between the energy landscape changes, effects on the brain state dynamics and causal behavioural responses using the mediation analysis.

### Replication experiment

We examined the replicability of the findings of the main experiments by repeating the main experiment with focusing on the seven conditions that yielded significant behavioural effects. This experiment employed a small but independent cohort (originally N = 15; N = 14 after one participant dropped out).

### Validation experiment with excitatory TMS

We also examined the validity of the main findings in experiments using the excitatory TMS (see details on the TMS protocol in 'Device setup: TMS' section). As shown in our previous work (*Watanabe et al., 2015*; *Watanabe et al., 2014a*), the excitatory TMS was expected to induce behavioural and neural effects opposite to those yielded by the inhibitory TMS. To test this, we repeated the main experiment using the excitatory TMS with an independent cohort (N = 15). Like the replication experiment, we focused on the seven TMS conditions that induced significant behavioural changes in the main experiment.

### Conventional quadripulse TMS experiment

For comparison, we examined the behavioural effects of conventional 30 min inhibitory quadripulse TMS (QPS) by employing the 14 individuals who completed the replication experiment at least 1 month before.

In this experiment, we adopted the inhibitory QPS protocols that were used in our previous studies (*Watanabe et al., 2015*; *Watanabe et al., 2014a*). This TMS consisted of 360 consecutive bursts with 5 s intervals (i.e. 30 min), and each burst comprised four monophasic 20 Hz TMS whose intensity was set at 90 % of AMT. We conducted this QPS over either of the three PFC regions using the same TMS device as used in the main experiment. In addition, we performed a sham condition; thus, the participants came to the lab four times with more-than-one-week interval. The order of the TMS conditions was randomised and balanced across the participants.

In each day, the participants first completed the six control runs of bistable visual perception tests and then underwent the 30 min TMS session over one of the three PFC sites. In the sham condition, the TMS coil was placed over one of the three PFC areas, which was randomly chosen and balanced across the participants. During the TMS session, they were asked to rest with their eyes open. Ten minutes after the end of the TMS, they took the six runs of the bistable visual perception tests.

### Longitudinal experiment

We investigated accumulative effects of the brain-sate-/history-specific TMS by a longitudinal TMS experiment that employed the 63 participants who completed the main, replication or validation experiment.

Like the replication and validation experiments, we focused on the seven TMS conditions that induced significant behavioural changes. In addition, we added three sham conditions, in which the TMS was administered over one of the three PFC regions at random timings. In total, the ten conditions were examined.

The 63 participants were randomly assigned to two of the ten conditions: each non-sham condition had 13 individuals, whereas two sham conditions had 12 participants and the other sham condition was examined with 11 individuals.

The accumulative effect of each TMS/sham condition was evaluated over nine weeks. In the first 8 weeks, the participants underwent weekly TMS/sham experiments. On each day, the participants underwent a TMS session (six runs of bistable visual perception tests) and two control sessions (three runs) before and after the TMS session. The details of the TMS protocol were the same as those for the main experiment. The brain site for the sham condition was randomly chosen from the three PFC areas for each participant for each session. The target brain sites were not changed during one period.

For each day, we analysed the behavioural effects of the state-/state-history-dependent TMS by calculating proportional changes in the median percept durations within each day. We also tracked

the behavioural effects over the 2-month experiment and examined whether the magnitude of the behavioural response at each day was significantly different from the first day (week 0).

In the last week (week 9), the participants underwent the control sessions only. Such baseline responses were used to evaluate whether the two-month weekly TMS affected the baseline perceptual stability.

## Validation of the specificity of the EEG recording

We examined whether the current EEG system had sufficient specificity to distinguish between the neighbouring prefrontal activities (i.e. the DLPFC activity and IFC activity).

First, using the EEG data in Control Experiments (N = 65), we tested whether the EEG signals recorded from the two prefrontal areas were significantly different from each other. Technically, we performed the offline energy landscape analysis after one of the prefrontal activities was deliberately replaced with the other activity. If this replacement would not disturb the original observations, it would be highly likely that the current EEG system could not detect the difference in the neural activity between the DLPFC and IFC. Inversely, if the original results were distorted, we could infer that the current EEG system has sufficient detectability of differences between the two prefrontal activities.

As a result, the significant brain-behaviour correlation seen in the original analysis ($r = 0.67$; *Figure 2j*) deteriorated when we set the DLPFC and IFC activities at the same values. If the IFC activity was replaced with the DLPFC activity, the correlation coefficient was reduced to 0.19 (*Figure 17a*). When we used the IFC activity for the DLPFC activity, the coefficient was decreased to 0.23 (*Figure 17b*). These results support the notion that the current EEG system can detect significant differences in neural activity between the two prefrontal regions.

Second, using the EEG data in the main experiment (N = 34), we compared neural responses to TMS between the DLPFC and IFC (*Figure 17c*). If our EEG system can have a sufficient SNR, the EEG signals recorded from the IFC would not be affected by TMS over DLPFC.

In fact, when we administered TMS over DLPFC, the DLPFC signal showed a significant reduction, but the IFC signal did not (*Figure 17d*). After TMS over IFC, the IFC signal was significantly decreased, whereas the DLPFC signal was not affected (*Figure 17e*).

In sum, these two analyses indicate sufficient spatial sensitivity and specificity in the current EEG system.

## Effects of microsaccade on EEG Recording

In the current study, we adopted a derivation method (i.e. Hjorth signal calculation) (*Keren et al., 2010*; *Pulvermüller et al., 1997*; *Trujillo et al., 2005*; *Zion-Golumbic et al., 2010*) and independent component analysis (ICA) (*Hassler et al., 2011*; *Jung et al., 2000*; *Lee et al., 1999*) to reduce the artefacts of microsaccades on gamma-band EEG signals (*Yuval-Greenberg et al., 2008*). We confirmed the effectiveness of these signal processing procedures in an additional EEG experiment employing 30 healthy individuals (age = 25.1 ± 2.1; female = 11).

In the experiment, the participants experienced the same bistable visual perception with EEG electrodes placed over slightly different places. Precisely, 28 electrodes were located around the same seven brain regions (ROIs) in the same manner as in the original experiment. The other four electrodes were placed around the eyes for electrooculography (EOG) in accordance with previous studies (*Croft and Barry, 2000*; *Elbert et al., 1985*; *Shan et al., 1995*): two EOG electrodes at the outer canthi of both the eyes (HEOGL and HEOGR) and two at the below and above the right eye (VEOGI and VEOGS). Using an electrode near Pz as a reference, we calculated a radial EOG (REOG) signal as follows: REOG = (HEOGR+ HEOGL + VEOGI+ VEOGS)/4 – Pz. This REOG signal was used to detect the timings of microsaccades in the same manner as shown in a line of previous studies (*Croft and Barry, 2000*; *Elbert et al., 1985*; *Keren et al., 2010*; *Shan et al., 1995*).

In parallel with this microsaccade detection, we applied the same preprocessing procedures to the EEG signals recorded from the ROIs.

Finally, we compared raw EEG signals with preprocessed signals around the timings of microsaccades and examined whether such preprocessing procedures could remove microsaccade-related artefacts. Technically, we compared the mean power between the microsaccade period ( ± 100 ms around a microsaccade) and peripheral period consisting of two 100 ms time windows around the microsaccade period (*Figure 18*).

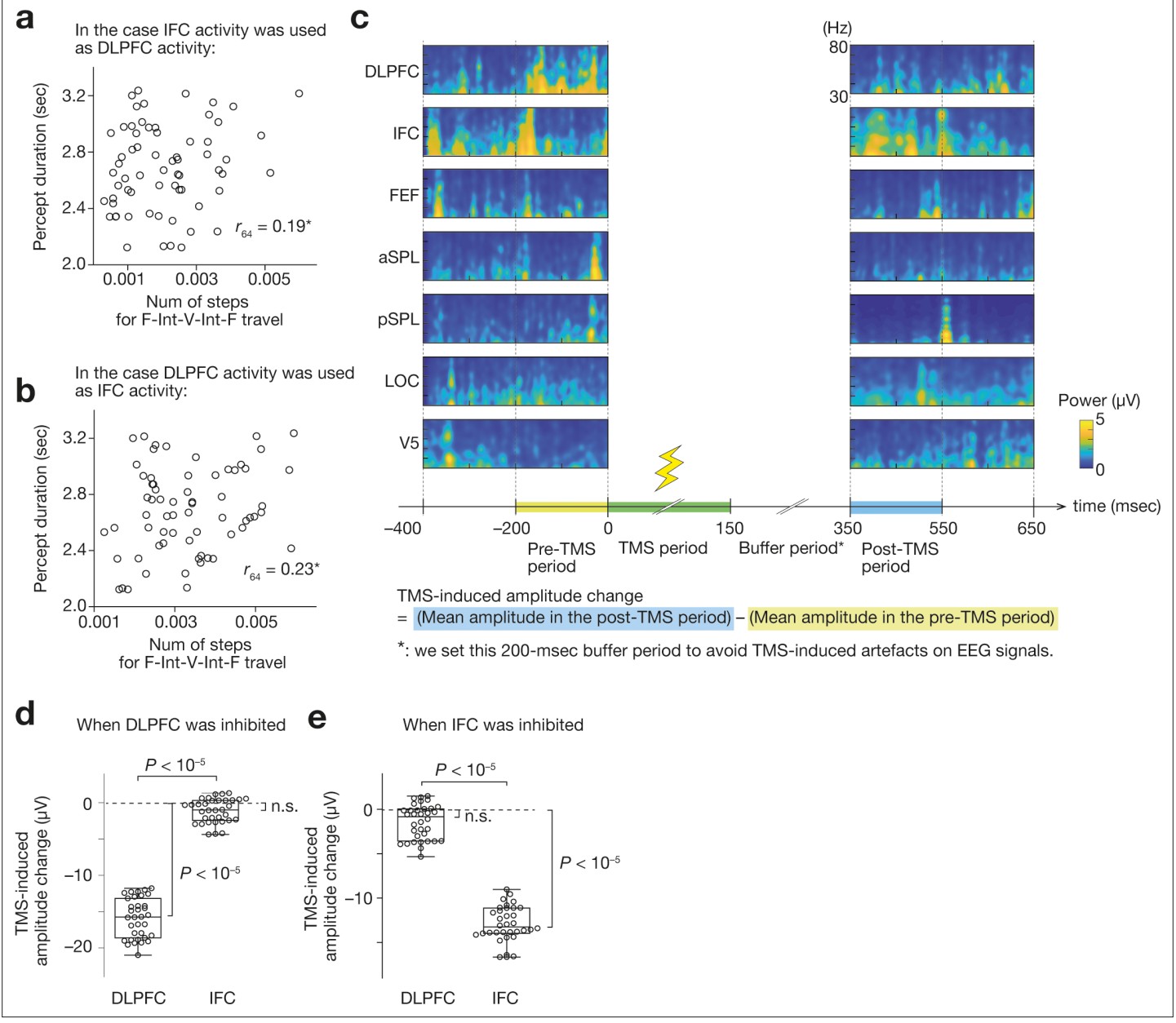

**Figure 17.** Spatial sensitivity and specificity of the current EEG system. (**a** and **b**) If we replaced one of the prefrontal activities with the other, the brain-behaviour association in the original result (**Figure 2j**) deteriorated. Panel a shows a result when the DLPFC activity was replaced with the IFC activity. Panel b shows a correlation when the IFC activity was set as the same value as the DLPFC activity. *, Z > 3.2, p < 0.0018 in the comparison of the brain-behaviour correlation between the original one and one after the operation to replace one prefrontal activity with the other. Each circle represents each participant. (**c**) We investigated the TMS-induced changes in the gamma-band amplitude. For example, the panel c shows the gamma-band signals for the seven ROIs before and after a TMS administration over the DLPFC in Participant 1. We calculated the mean amplitude of the gamma-band signals in both pre-TMS and post-TMS periods and estimated the neural effects of the TMS. (**d** and **e**) When we applied TMS over the DLPFC, the DLPFC activity was significantly reduced but the IFC activity was not (panel d). When the TMS was performed over the IFC, only the IFC activity was inhibited (panel e). Each circle represents each participant.

As a result, we found that both the derivation method and ICA reduced the microsaccade-relevant artefacts. The derivation substantially removed the signal increase around a microsaccade (e.g. see **Figure 18b** for effects on DLPFC signal), which was confirmed in all the ROI signals quantitatively (**Figure 18c**). That is, for all the seven ROIs, the mean power in the microsaccade period was not significantly different from that in the peripheral period (p > 0.05 in one-sample *t*-tests).

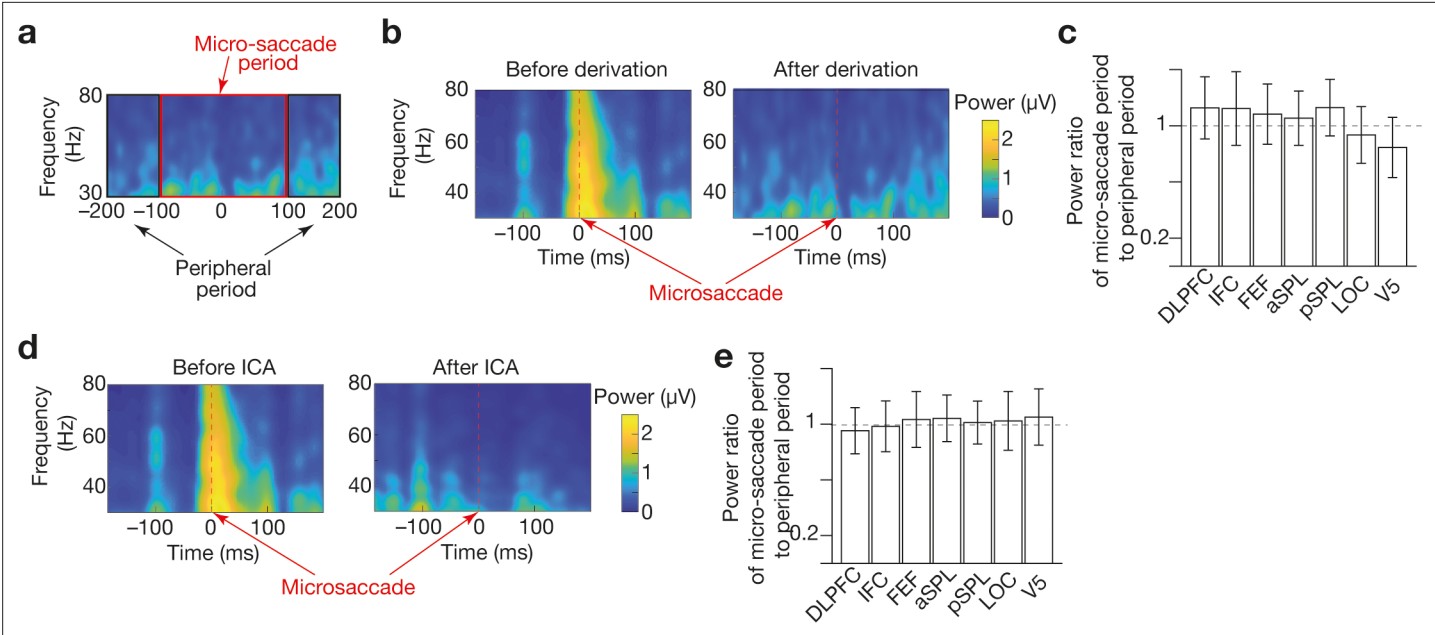

**Figure 18.** Preprocessing to reduce microsaccade-relevant effects on EEG data . We adopted a derivation method (*Keren et al., 2010*; *Pulvermüller et al., 1997*; *Trujillo et al., 2005*; *Zion-Golumbic et al., 2010*) and independent component analysis (ICA) (*Hassler et al., 2011*; *Jung et al., 2000*; *Lee et al., 1999*) to reduce the artefacts of microsaccades on gamma-band EEG signals. To evaluate the effectiveness of these methods, we compared the mean amplitudes of the gamma-band signals between the microsaccade and peripheral period (**a**). The timings of microsaccades were determined based on EOG signals (*Croft and Barry, 2000*; *Elbert et al., 1985*; *Keren et al., 2010*; *Shan et al., 1995*). The derivation method was able to reduce the artefacts of the microsaccades (panel **b** for an example of IFC signals in Participant 1) in all the seven regions of interest (ROIs; panel **c**). The ICA could also substantially weaken such effects (panel **d** for an example of IFC signals in Participant 1) in all the seven ROIs (**e**).

The ICA was effective as well. The procedure reduced the signal increase induced by a microsaccade (*Figure 18d*) in all the ROIs (p > 0.05 in one-sample *t*-tests; *Figure 18e*).

## Statistics

In the case of multiple comparisons, basically, we conducted a two-way ANOVA (participant× condition) and post-hoc *t*-tests with Bonferroni correction.

## Acknowledgements

This work was supported by Grant-in-aid for Research Activity from Japan Society for Promotion of Sciences, The University of Tokyo Excellent Young Researcher Project, Fukuhara Foundation, Yamaha Motor Foundation for Sports, Astellas Foundation for Research on Metabolic Disorders, Showa University Medical Institute of Developmental Disabilities Research and JST Moonshot R&D Program (JPMJMS2021). Accessibility to some experimental facilities used here was realised through financial support from Yasushi Miyashita's team in RIKEN CBS, Japan. The author appreciates the stimulating hints and sharp insights of Professor Geraint Rees (UCL, UK) in the initial phase of this research.

## Additional information

### Funding

| Funder | Grant reference number | Author |
|---|---|---|
| Japan Society for the Promotion of Science | 19H03535 | Takamitsu Watanabe |
| Japan Science and Technology Agency | JPMJMS2021 | Takamitsu Watanabe |

| Funder | Grant reference number | Author |
|---|---|---|
| Astellas Foundation for Research on Metabolic Disorders | | Takamitsu Watanabe |
| Fukuhara Foundation | | Takamitsu Watanabe |
| Yamaha Motor Foundation of Sports | | Takamitsu Watanabe |
| Showa University Medical Institute of Developmental Disabilities Research | | Takamitsu Watanabe |
| The University of Tokyo Excellent Young Researcher Project | | Takamitsu Watanabe |

The funders had no role in study design, data collection and interpretation, or the decision to submit the work for publication.

## Author contributions

Takamitsu Watanabe, Conceptualization, Data curation, Formal analysis, Funding acquisition, Investigation, Methodology, Project administration, Software, Validation, Visualization, Writing - original draft, Writing - review and editing

## Author ORCIDs

Takamitsu Watanabe (iD) http://orcid.org/0000-0002-8104-6873

## Ethics

This study was approved by Institutional Ethics Committees in RIKEN and The University of Tokyo. The TMS protocols used here complied with the guideline issued by the Japanese Society for Clinical Neurophysiology and that by International Federation of Clinical Neurophysiology. All the participants provided written informed consents before any experiment and were financially compensated for their participation.

## Decision letter and Author response

Decision letter https://doi.org/10.7554/eLife.69079.sa1
Author response https://doi.org/10.7554/eLife.69079.sa2

---

# Additional files

## Supplementary files

• Transparent reporting form

## Data availability

The behavioural data are deposited in Dryrad (https://doi.org/10.5061/dryad.8931zcrqn) and the codes for the energy landscape analysis has been shared as a supplementary information of our previous work (Ezaki, T., Watanabe, T., Ohzeki, M. & Masuda, N. Energy landscape analysis of neuroimaging data. Philosophical Transactions Royal Soc Math Phys Eng Sci 375, 20160287 (2017), https://doi.org/10.1098/rsta.2016.0287).

The following dataset was generated:

| Author(s) | Year | Dataset title | Dataset URL | Database and Identifier |
|---|---|---|---|---|
| Watanabe T | 2021 | Data from: Causal roles of prefrontal cortex during spontaneous perceptual switching are determined by brain state dynamics | https://doi.org/10.5061/dryad.8931zcrqn | Dryad Digital Repository, 10.5061/dryad.8931zcrqn |

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
