## [Editor Report]

This single-author work, by combining real-time closed-loop EEG-TMS and sophisticated computational modelling to characterize ongoing brain states, impressively demonstrates the causal role and different functions of several prefrontal regions in modulating bistable perception, in a brain-state-dependent way.

---

## [Decision Letter]

**Decision letter after peer review:**

Thank you for submitting your article "Causal roles of prefrontal cortex during spontaneous perceptual switching are determined by brain state dynamics" for consideration by *eLife*. Your article has been reviewed by 3 peer reviewers, and the evaluation has been overseen by a Reviewing Editor and Chris Baker as the Senior Editor. The following individual involved in review of your submission has agreed to reveal their identity: Tomas Knapen (Reviewer #1).

Essential revisions:

(1) The study is based on findings and methods in the author's previous fMRI work. However, since the current work employs EEG recording, it is important to add several validation and control analyses or data, including: (i) replication of the relationship between individual brain state dynamics and.bistable perception; (ii) providing data supporting that the EEG-channel-based TMS has high enough SNR to differentiate between neighboring regions in prefrontal cortex; (iii) fleshing out the numerical simulation procedure used to calculate dwelling time.

(2) Since the state transition in EEG signals is defined in terms of γ-band power, the authors should: (i) provide control data to exclude the involvement of microsaccade-related artifacts that might elicit the γ-band response, especially considering the structure-from-motion stimulus used here ; (ii) present time-frequency plots before/after state transition as well as TMS-triggered time-frequency plots to help readers have a transparent understanding of the state transition and TMS effects.

(3) Addressing the role of attention, i.e., is the effect of DLPFC stimulation on the dynamics of bistable perception mediated by changes in attentional state?

(4) Addressing the possible circular hypothesis testing (Page 10). Specifically, the models were fit by using EEG data (and behavior?) to calculate the energy landscape, so is it trivially expected that the dwell times seen behaviorally correlate with the energy barrier estimated by the model?

(5) Energy landscape is a very abstract term and needs to be fleshed out in terms of the present results, e.g., specifying functional roles of different regions of prefrontal cortex in bistable perception.

*Reviewer #1 (Recommendations for the authors):*

I find it hard to understand the impact statement, title, and abstract. A naive reader will likely find these sentences very hard to parse. A reader will likely wonder; What are the different causal roles of prefrontal cortex that are changing during bistable perception? What are hypothetical energy landscapes?

Abstract; presumable is overly vague; why not just presumed? Abstract sentences are also very long, this taxes the reader.

The initial paragraph of the results requires the reader to either be an expert in the author's previous energy landscape work, or take a lot of things at face value. I suggest this way of writing is going way to fast. I appreciate that the author wants to get to the 'meat' of the study, but this requires much more explanation.

*Reviewer #2 (Recommendations for the authors):*

An impressive study. My main suggestions are mostly stated in the public comments. Basically I'd like the author to explicitly discuss:

1. The role of attention. Is the effect of DLPFC stimulation on the dynamics of bistable perception mediated by changes in attentional state?

2. What are the functional roles of the different regions of DLPFC in bistable perception? Energy landscape is rather abstract. The author should help readers to gain some intuition about the link between brain states and perceptual states.

3. Reportability of perceptual states was proposed as an important factor in observing the PFC engagement in bistable perception. The author should discuss the implication of the current study on the significance of perceptual report for PFC's engagement in bistable perception.

4. The descriptions in the paper on "binocular rivalry".. "has been often explained by …" (pages 3 and 14) are inaccurate, should be revised.

*Reviewer #3 (Recommendations for the authors):*

1) A recent study (Weilnhammer et al., Curr Bio, 2021) showed that theta-burst TMS over IFC (same as pDLPFC here) prolonged percept duration. By contrast, continuous TMS here had null-effect (Figure 2c), and single-pulse inhibitory TMS reduced percept duration in the F-state (i.e., opposite to the Weilnhammer et al., finding). Since the Weilnhammer et al., paper was published after this manuscript was submitted (even though a bioRxiv version was available), it is not obligatory for the author to cite this paper. Nonetheless, in the revision, some discussion about potential sources of this discrepancy would be helpful.

2) Introduction, 1st paragraph states that the PFC has in particular been thought to be involved in spontaneous switching. But I think the evidence for a causal involvement in spontaneous switching is actually stronger in parietal cortex, given the long series of TMS studies there by Kanai and Rees.

3) Figure 5. How excitatory vs. inhibitory TMS was conducted should be described in the main text.

4) Results (Figure 5-6) should be described in the Results section, not Discussion.

5) The description of the maximum entropy model is rather inaccessible. Breaking it down, providing intuitions and code, and intermediate steps in intuitive/graphic language, would be helpful.

---

## [Author Response]

Essential revisions:(1) The study is based on findings and methods in the author's previous fMRI work. However, since the current work employs EEG recording, it is important to add several validation and control analyses or data, including: (i) replication of the relationship between individual brain state dynamics and.bistable perception; (ii) providing data supporting that the EEG-channel-based TMS has high enough SNR to differentiate between neighboring regions in prefrontal cortex; (iii) fleshing out the numerical simulation procedure used to calculate dwelling time.(i) replication of the relationship between individual brain state dynamics and bistable perception

We are sorry for the insufficient descriptions on the applicability of the energy landscape analysis to the current EEG data. To address this, we have added the following descriptions that demonstrate that the current EEG-based energy landscape analysis was accurate enough to replicate our previous fMRI-based observation at both the group and individual levels.

Results section:

“Validation of energy landscape analysis in EEG

As a preparation, we examined whether the current EEG system could capture qualitatively the same brain state dynamics as those found in our previous fMRI study employing the same SFM stimulus^1^. […] In sum, these results show that, at least with an offline analysis, the current EEG system can identify qualitatively the same brain state dynamics underpinning the SFM-induced bistable visual perception as seen in our previous fMRI study^1^.”

ii) providing data supporting that the EEG-channel-based TMS has high enough SNR to differentiate between neighboring regions in prefrontal cortex

We agree on the importance of such control data to support the detectability and specificity of our EEG system. To clarify that the current system had high enough SNR to distinguish between neural signals of the two neighbouring prefrontal regions (i.e., DLPFC and IFC), we performed the following two new analyses.

First, we confirmed that the EEG signals recorded from the DLPFC and IFC were significantly different from each other. Technically, using the EEG data in the Control Experiment, we examined what would happen to the brain-behaviour association (Figure 1m) if we deliberately made both the prefrontal activities the same. If the current system cannot differentiate between the DLPFC and IFC activities, the neural signals recorded from the DLPFC and IFC should be almost the same; thus, the significant brain-behaviour association, which is observed in the original analysis, should not be affected even after the DLPFC and IFC activities are set at the same value.

In reality, however, such an operation disturbed the brain-behaviour correlation. When we replaced the DLPFC activity with the IFC activity and conducted the entire offline energy landscape analysis, the resultant length of the F-Int-V-Int-F travel was not so strongly correlated with the percept duration (*r* = 0.19; Figure 17a) as in the original analysis (*r* = 0.67). When we relaced the IFC activity with the DLPFC activity, the correlation was also weakened (*r* = 0.23; Figure 17b).

These results demonstrate that the EEG signals obtained from the electrodes placed on the DLPFC and IFC were significantly different from each other.

Second, we confirmed the sufficiently high spatial specificity of this system by examining neural responses to TMS over the prefrontal areas (Figure 17c). When we administered the TMS over the DLPFC electrode, the EEG signals recorded from DLPFC was significantly reduced, whereas the signals of IFC were almost intact (Figure 17d). When the TMS was applied over the IFC, the IFC signal was decreased whilst the DLPFC activity was not (Figure 17e).

In sum, these new observations clarified that the current system was able to distinguish between the neighbouring prefrontal regions.

To clarify this, we have now added the following descriptions and a new figure (Figure 17).

Methods section:

“5. Validation of the specificity of the EEG recording.

We examined whether the current EEG system had sufficient specificity to distinguish between the neighbouring prefrontal activities (i.e., the DLPFC activity and IFC activity). […] In sum, these two analyses indicate sufficient spatial sensitivity and specificity in the current EEG system.”

iii) fleshing out the numerical simulation procedure used to calculate dwelling time.

We are sorry for the unclear description of the numerical simulation procedure. Now in accordance with the suggestion, we have re-written the methods as follows:

Methods section:

“3.2.5. Offline EEG analysis: simulation of brain state dynamics

In the final part of the energy landscape analysis, we probed the brain state dynamics by a random-walk simulation on the energy landscape ^1,2^. […] Finally, we counted how long each of the major brain states continued in the brain state trajectory (dwelling time) and how often one major brain state transited to another major state (transition frequency).”

(2) Since the state transition in EEG signals is defined in terms of γ-band power, the authors should: (i) provide control data to exclude the involvement of microsaccade-related artifacts that might elicit the γ-band response, especially considering the structure-from-motion stimulus used here ; (ii) present time-frequency plots before/after state transition as well as TMS-triggered time-frequency plots to help readers have a transparent understanding of the state transition and TMS effects.(i) provide control data to exclude the involvement of microsaccade-related artifacts that might elicit the γ-band response, especially considering the structure-from-motion stimulus used here

We agree on the necessity of reducing the microsaccade-related artefacts on the γ-band signals. In fact, we adopted a derivation method (i.e., Hjorth signal calculation) and ICA to reduce such artefacts (for the derivation method, see ^5–8^; for the ICA, see ^9–11^).

In the meantime, we admit that the original manuscript did not present explicit results to support effects of these signal processing methods.

To address this situation, we conducted a new EEG experiment, in which 30 healthy adults underwent the same psychophysics paradigm. In the additional experiment, 28 EEG electrodes were placed around the seven regions of interests (ROIs) in the same manner as in the original experiment, whereas the other four electrodes were located around the eyes for electrooculography (EOG)^5^. Based on previous literature^12–14^, we used these EOG signals to infer the timings of the occurrences of microsaccades.

We then examined how each of the pre-processing procedures changes EEG signals around each microsaccade. As a result, both the derivation method and ICA significantly reduced the microsaccade-related artefacts. For example, a microsaccade induced a large γ-band activity in the electrode over the IFC (left panels of Figures 18b and 18d), which was supressed by the derivation method (right panel of Figure 18b) and ICA (right panel of Figure 18d). This tendency was confirmed in the electrodes over the other ROIs (Figures 18c and 18e) by comparing signal amplitude between the microsaccade period and peripheral one (Figure 18a).

These results support that the current pre-processing procedures can sufficiently reduce the microsaccade-related artefacts. We clarified these issues by adding the following descriptions and new figure (Figure 18) into the Methods section.

Methods section:

“4. Effects of microsaccade on EEG recording

In the current study, we adopted a derivation method (i.e., Hjorth signal calculation)^85–88^ and independent component analysis (ICA)^89–91^ to reduce the artefacts of microsaccades on γ-band EEG signals^92^. […] The ICA was effective as well. The procedure reduced the signal increase induced by a microsaccade (Figure 18d) in all the ROIs (*P* > 0.05 in one-sample *t*-tests; Figure 18e).”

ii) present time-frequency plots before/after state transition as well as TMS-triggered time-frequency plots to help readers have a transparent understanding of the state transition and TMS effects.

In accordance with the suggestion, we have now added the time-frequency plots showing the transitions of brain state (Figure 13e) and those showing effects of TMS (Figure 17c).

(3) Addressing the role of attention, i.e., is the effect of DLPFC stimulation on the dynamics of bistable perception mediated by changes in attentional state?

We admit that this study cannot directly examine associations between attention and DLPFC activity in the brain state dynamics because we monitored no attention-related biological or behavioural metrics during the bistable perception. In the meantime, based on the current findings and previous literature, we can infer the role of attention in this finding. Therefore, now, in accordance with the suggestion, we have added the following discussions on this issue:

Discussion section:

“Moreover, the current findings suggest distinct functions of the PFC regions in terms of the brain state dynamics: the activation of DLPFC enhances the functional integration between the Frontal and Intermediate state, whereas the IFC activity promotes the functional segregation between the two brain states; the FEF activity stabilises Frontal state. […] Given these, it would be more reasonable to infer that the top-down and bottom-up signals, which are supposed to be generated and communicated in the brain state dynamics, contain not only attention-related information but also other types of neural signal such as prediction error in predictive coding paradigm^19–22^.”

(4) Addressing the possible circular hypothesis testing (Page 10). Specifically, the models were fit by using EEG data (and behavior?) to calculate the energy landscape, so is it trivially expected that the dwell times seen behaviorally correlate with the energy barrier estimated by the model?

We are sorry for our insufficient description on the issue. As in our previous fMRI study^1^, the current energy landscape analysis used no behavioural information, such as the timings of perceptual switching, to identify and quantify the brain state dynamics. Therefore, the brain-behaviour correlations seen in this study are not consequences of tautology but evidence that the analysis captured brain dynamics underpinning the bistable visual perception.

To clarify this, we added the following a sentence in the Results sections and Methods section:

Results section:

“Note that these energy landscape analyses used no behavioural information to identify the brain state dynamics; thus, the significant brain-behaviour correlations (Figures 2j-2m) are not consequences of circular analysis.”

Methods section:

“Note that we used no behavioural response, such as the timing of perceptual switch, in the following preprocessing procedure and energy landscape analysis.”

(5) Energy landscape is a very abstract term and needs to be fleshed out in terms of the present results, e.g., specifying functional roles of different regions of prefrontal cortex in bistable perception.

We are sorry for our unclear descriptions on the functional roles of the prefrontal regions in the bistable visual perception. To clarify the functions of the prefrontal areas, we have added the following interpretations of the current findings into the Discussion section:

Discussion section:

“Moreover, the current findings suggest distinct functions of the PFC regions in terms of the brain state dynamics: the activation of DLPFC enhances the functional integration between the Frontal and Intermediate state, whereas the IFC activity promotes the functional segregation between the two brain states; the FEF activity stabilises Frontal state. […] The activation of FEF can be regard as a support for the top-down signal generation.”

We appreciate for the reviewer's helpful and constructive comments which we believe have substantially improved our manuscript.

Reviewer #1 (Recommendations for the authors):I find it hard to understand the impact statement, title, and abstract. A naive reader will likely find these sentences very hard to parse. A reader will likely wonder; What are the different causal roles of prefrontal cortex that are changing during bistable perception? What are hypothetical energy landscapes?Abstract; presumable is overly vague; why not just presumed? Abstract sentences are also very long, this taxes the reader.

We greatly appreciate for the reviewer's expressing such frank concerns on our manuscript. Due to the word limitation, we cannot fully explain the details of the methods, but we have modified the impact statement and abstract to address the concerns at the most.

Abstract:

“The prefrontal cortex (PFC) is thought to orchestrate cognitive dynamics. However, in tests of bistable visual perception, no direct evidence supporting such presumable causal roles of the PFC has been reported. […] This work resolves the controversy over the PFC roles in spontaneous perceptual switching and underlines brain state dynamics in fine investigations of brain-behaviour causality.”

Impact statement

“Prefrontal causal roles in bistable perception are dynamically changing and determined by the brain state to which the whole-brain activity pattern belongs.”

The initial paragraph of the results requires the reader to either be an expert in the author's previous energy landscape work, or take a lot of things at face value. I suggest this way of writing is going way to fast. I appreciate that the author wants to get to the 'meat' of the study, but this requires much more explanation.

We are sorry for such inconvenience and extra efforts that we asked to the reviewer. Now we have added more details into the first part of the Results section. Please see our response to (1)-(i) in Responses to the essential revisions.

We appreciate for the reviewer's helpful and constructive comments which we believe have substantially improved our manuscript.

Reviewer #2 (Recommendations for the authors):An impressive study. My main suggestions are mostly stated in the public comments. Basically I'd like the author to explicitly discuss:1. The role of attention. Is the effect of DLPFC stimulation on the dynamics of bistable perception mediated by changes in attentional state?

We have now discussed this issue as stated above. We hope they work.

2. What are the functional roles of the different regions of DLPFC in bistable perception? Energy landscape is rather abstract. The author should help readers to gain some intuition about the link between brain states and perceptual states.

We admit that the energy-landscape-based notion appears to be quite abstract. Although we have to be careful to directly link the current observations to some conventional psychological framework, we have now stated neuropsychological interpretation of the findings in the Discussion section. Please find them in our response to (5) in Responses to the essential revision.

3. Reportability of perceptual states was proposed as an important factor in observing the PFC engagement in bistable perception. The author should discuss the implication of the current study on the significance of perceptual report for PFC's engagement in bistable perception.

We agree on the importance of the reportability of the perceptual states in the bistable perception tests. Now, in accordance with the reviewer's suggestion, we have discussed this issue as follows:

Discussion section:

“By the same logic, it may be difficult to regard the current findings as evidence supporting the notion that the DLPFC and IFC are irrelevant to the bistable perception itself but only involved in reporting the perceptual states^16,18^. […] To resolve this situation, future studies would have to examine brain-state-dependent behavioural causality of the PFC in SFM-induced bistable perception using the non-report paradigm^46^.”

4. The descriptions in the paper on "binocular rivalry".. "has been often explained by …" (pages 3 and 14) are inaccurate, should be revised.

We have now revised them as follows:

Introduction section:

“To focus on neural mechanisms in the higher-order cortex, we did not adopt a test of binocular rivalry, in which the fluctuating percept has often been linked with the neural activity in the lower-level brain systems such as visual cortex^32–38^.”

Discussion section:

“These findings may not be directly applicable to other types of multistable visual perception, such as binocular rivalry, which is linked to lower-level brain architectures such as the visual cortex ^32–38^. […] Given this, the current observations might be more applicable to types of bistable perception that requires construction of a 3D image from 2D motion compared to the other types such as the binocular rivalry.”

We appreciate for the reviewer's helpful and constructive comments which we believe have substantially improved our manuscript.

Reviewer #3 (Recommendations for the authors):1) A recent study (Weilnhammer et al., Curr Bio, 2021) showed that theta-burst TMS over IFC (same as pDLPFC here) prolonged percept duration. By contrast, continuous TMS here had null-effect (Figure 2c), and single-pulse inhibitory TMS reduced percept duration in the F-state (i.e., opposite to the Weilnhammer et al. finding). Since the Weilnhammer et al. paper was published after this manuscript was submitted (even though a bioRxiv version was available), it is not obligatory for the author to cite this paper. Nonetheless, in the revision, some discussion about potential sources of this discrepancy would be helpful.

In accordance with the reviewer's suggestion, we have added the following paragraph to discuss this issue:

Discussion section:

“If this is the case, why could the recent study successfully detect prolonged percept in the bistable visual perception test after applying conventional TMS over the IFC?^13^ The TMS protocols are different between the previous study and the current work, and further studies are necessary to answer this question directly; but we can speculate the reason as follows. In the recent study, theta-burst TMS was administered to the IFC during rest. Considering that default-mode network is mainly active during rest and the IFC and DLPFC tend to be inactive, we can speculate that the TMS was applied during brain states similar to Visual state. As shown in Figure 4b, such V-state-dependent TMS over the IFC could induce a moderate prolongation of the percept duration (Cohen's *d* = 0.3), which may be detected as a significant behavioural effect in the recent study.”

2) Introduction, 1st paragraph states that the PFC has in particular been thought to be involved in spontaneous switching. But I think the evidence for a causal involvement in spontaneous switching is actually stronger in parietal cortex, given the long series of TMS studies there by Kanai and Rees.

We agree on the point. In the meantime, the goal of the current study is to examine the prefrontal causality in the bistable perception. Therefore, to make it simple and clear, the original manuscript did not refer to the previous TMS studies on the parietal cortex.

Now we have added the following sentences to clarify such parietal roles in this cognitive task:

Discussion section:

“In the bistable visual perception paradigm, a line of previous TMS studies reported behavioural causal roles of the parietal cortex in the right hemisphere^46–50^, whereas no investigation found such effects in the prefrontal cortex except for a recent one^13^.”

3) Figure 5. How excitatory vs. inhibitory TMS was conducted should be described in the main text.

In accordance with the reviewer's suggestion, we have now stated the results of the experiment using excitatory TMS in the Results sections:

Results section:

“Moreover, another independent experiment (*N*=15) showed that, as in our previous work ^40,41^, the excitatory TMS induced behavioural effects opposite to those yielded by the inhibitory stimulation (*t*_14_>2.8, *P*<0.01 in one-sample *t*-tests; Figure 9b), which added indirect but empirical support for the current observations.”

4) Results (Figure 5-6) should be described in the Results section, not Discussion.

In accordance with the reviewer's suggestion, we have moved them into the Results section.

5) The description of the maximum entropy model is rather inaccessible. Breaking it down, providing intuitions and code, and intermediate steps in intuitive/graphic language, would be helpful.

To address the reviewer's concern, we have entirely re-written the Methods sections on the maximum entropy model as follows:

Methods section:

“We then conducted the energy landscape analysis ^20,24–27^ of the preprocessed datasets of the seven ROIs. […] Based on this definition, we adjusted *h_i_* and *J_ij_* until these the ⟨σi⟩m and ⟨σiσj⟩m were approximately equal to the empirically obtained ⟨σi⟩ and ⟨σiσj⟩ using a gradient ascent algorithm.”

We appreciate for the reviewer's helpful and constructive comments which we believe have substantially improved our manuscript.

References

1. Watanabe, T., Masuda, N., Megumi, F., Kanai, R. and Rees, G. Energy landscape and dynamics of brain activity during human bistable perception. *Nat Commun* 5, 4765 (2014).

2. Watanabe, T. and Rees, G. Brain network dynamics in high-functioning individuals with autism. *Nat Commun* 8, 16048 (2017).

3. Massen, C. P. and Doye, J. P. K. Identifying communities within energy landscapes. *Phys Rev E* 71, 046101 (2005).

4. Girvan, M. and Newman, M. E. J. Community structure in social and biological networks. *Proc National Acad Sci* 99, 7821–7826 (2002).

5. Keren, A. S., Yuval-Greenberg, S. and Deouell, L. Y. Saccadic spike potentials in γ-band EEG: Characterisation, detection and suppression. *Neuroimage* 49, 2248–2263 (2010).

6. Pulvermüller, F., Birbaumer, N., Lutzenberger, W. and Mohr, B. High-frequency brain activity: Its possible role in attention, perception and language processing. *Prog Neurobiol* 52, 427–445 (1997).

7. Trujillo, L. T., Peterson, M. A., Kaszniak, A. W. and Allen, J. J. B. EEG phase synchrony differences across visual perception conditions may depend on recording and analysis methods. *Clin Neurophysiol* 116, 172–189 (2005).

8. Zion-Golumbic, E., Kutas, M. and Bentin, S. Neural Dynamics Associated with Semantic and Episodic Memory for Faces: Evidence from Multiple Frequency Bands. *J Cognitive Neurosci* 22, 263–277 (2010).

9. Hassler, U., Barreto, N. T. and Gruber, T. Induced γ band responses in human EEG after the control of miniature saccadic artifacts. *Neuroimage* 57, 1411–1421 (2011).

10. Jung, T. *et al.* Removing electroencephalographic artifacts by blind source separation. *Psychophysiology* 37, 163–178 (2000).

11. Lee, T.-W., Girolami, M. and Sejnowski, T. J. Independent Component Analysis Using an Extended Infomax Algorithm for Mixed Subgaussian and Supergaussian Sources. *Neural Comput* 11, 417–441 (1999).

12. Croft, R. J. and Barry, R. J. Removal of ocular artifact from the EEG: a review. *Neurophysiologie Clinique Clin Neurophysiology* 30, 5–19 (2000).

13. Elbert, T., Lutzenberger, W., Rockstroh, B. and Birbaumer, N. Removal of ocular artifacts from the EEG — A biophysical approach to the EOG. *Electroen Clin Neuro* 60, 455–463 (1985).

14. Shan, Y., Moster, M. L. and Roemer, R. A. The effects of time point alignment on the amplitude of averaged orbital presaccadic spike potential (SP). *Electroen Clin Neuro* 95, 475–477 (1995).

15. Yuval-Greenberg, S., Tomer, O., Keren, A. S., Nelken, I. and Deouell, L. Y. Transient Induced Γ-Band Response in EEG as a Manifestation of Miniature Saccades. *Neuron* 58, 429–441 (2008).

16. Corbetta, M., Patel, G. and Shulman, G. L. The Reorienting System of the Human Brain: From Environment to Theory of Mind. *Neuron* 58, 306–324 (2008).

17. Corbetta, M. and Shulman, G. L. Control of goal-directed and stimulus-driven attention in the brain. *Nat Rev Neurosci* 3, 201–215 (2002).

18. Baldauf, D. and Desimone, R. Neural Mechanisms of Object-Based Attention. *Science* 344, 424–427 (2014).

19. Weilnhammer, V. *et al.* An active role of inferior frontal cortex in conscious experience. *Curr Biol* 31, 2868-2880.e8 (2021).

20. Brascamp, J., Sterzer, P., Blake, R. and Knapen, T. Multistable Perception and the Role of Frontoparietal Cortex in Perceptual Inference. *Annu Rev Psychol* 69, 1–27 (2017).

21. Weilnhammer, V., Stuke, H., Hesselmann, G., Sterzer, P. and Schmack, K. A predictive coding account of bistable perception – a model-based fMRI study. *Plos Comput Biol* 13, e1005536 (2017).

22. Hohwy, J., Roepstorff, A. and Friston, K. Predictive coding explains binocular rivalry: An epistemological review. *Cognition* 108, 687–701 (2008).

23. Sterzer, P., Russ, M. O., Preibisch, C. and Kleinschmidt, A. Neural Correlates of Spontaneous Direction Reversals in Ambiguous Apparent Visual Motion. *Neuroimage* 15, 908–916 (2002).

24. Knapen, T., Brascamp, J., Pearson, J., Ee, R. van and Blake, R. The Role of Frontal and Parietal Brain Areas in Bistable Perception. *J Neurosci* 31, 10293–10301 (2011).

25. Kleinschmidt, A., Bchel, C., Zeki, S. and Frackowiak, R. S. J. Human brain activity during spontaneously reversing perception of ambiguous figures. *Proc Royal Soc Lond Ser B Biological Sci* 265, 2427–2433 (1998).

26. Kanai, R., Bahrami, B. and Rees, G. Human Parietal Cortex Structure Predicts Individual Differences in Perceptual Rivalry. *Curr Biol* 20, 1626–1630 (2010).

27. Kanai, R., Carmel, D., Bahrami, B. and Rees, G. Structural and functional fractionation of right superior parietal cortex in bistable perception. *Curr Biol* 21, R106–R107 (2011).

28. Carmel, D., Walsh, V., Lavie, N. and Rees, G. Right parietal TMS shortens dominance durations in binocular rivalry. *Curr Biol* 20, R799–R800 (2010).

29. Freeman, E. D., Sterzer, P. and Driver, J. fMRI correlates of subjective reversals in ambiguous structure-from-motion. *J Vision* 12, 35–35 (2012).

30. Ezaki, T., Watanabe, T., Ohzeki, M. and Masuda, N. Energy landscape analysis of neuroimaging data. Philosophical Transactions Royal Soc Math Phys Eng Sci 375, 20160287 (2017).

31. Ezaki, T., Sakaki, M., Watanabe, T. and Masuda, N. Age‐related changes in the ease of dynamical transitions in human brain activity. *Hum Brain Mapp* 39, 2673–2688 (2018).

32. Vidaurre, D. *et al.* Discovering dynamic brain networks from big data in rest and task. *Neuroimage* 180, 646–656 (2018).

33. Baker, A. P. *et al.* Fast transient networks in spontaneous human brain activity. *eLife* 3, e01867 (2014).

34. Ezaki, T., Himeno, Y., Watanabe, T. and Masuda, N. Modelling state‐transition dynamics in resting‐state brain signals by the hidden Markov and Gaussian mixture models. *Eur J Neurosci* 54, 5404–5416 (2021).

35. Miller, P. and Katz, D. B. Stochastic Transitions between Neural States in Taste Processing and Decision-Making. *J Neurosci* 30, 2559–2570 (2010).